# SLC38A9 is directly involved in Tat-induced endolysosome dysfunction and senescence in astrocytes

Neda Rezagholizadeh, Gaurav Datta, Wendie A Hasler, Erica C Nguon, Elise V Smokey, Nabab Khan, Xuesong Chen

**Cellular senescence contributes to accelerated aging and the development of various neurodegeneration disorders including HIV-associated neurocognitive disorders. The development of HIV-associated neurocognitive disorders is attributed, at least in part, to the CNS persistence of HIV-1 transactivator of transcription (Tat), an essential protein for viral transcription that is actively secreted from HIV-1–infected cells. Secreted Tat enters cells via receptor-mediated endocytosis and induces endolysosome dysfunction and cellular senescence in CNS cells. Given that endolysosome dysfunction represents an early step in exogenous Tat-induced cellular senescence, we tested the hypothesis that Tat induces cellular senescence via an endolysosome-dependent mechanism in human astrocytes. We demonstrated that internalized Tat interacts with an endolysosome-resident arginine sensor SLC38A9 via the arginine-rich basic domain. Such an interaction between Tat and SLC38A9 leads to endolysosome dysfunction, enhanced HIV-1 LTR transactivation, and cellular senescence. These findings suggest that endolysosome dysfunction drives the development of senescence and highlight the novel role of SLC38A9 in Tat-induced endolysosome dysfunction and astrocyte senescence.**

## Introduction

Cellular senescence is a state of stable cell cycle arrest with secretory features in response to various cellular stress (Hernandez-Segura et al, 2018; Gonzalez-Gualda et al, 2021). A prominent feature of cellular senescence is the emergence of senescence-associated secretory phenotype (SASP), in which process cytokines, chemokines, matrix remodeling proteins, and growth factors are secreted into the tissue microenvironment (Gorgoulis et al, 2019). Besides the direct degenerating nature of cellular senescence (Novais et al, 2021), pro-inflammatory and pro-oxidative factors secreted by senescent cells could elicit deleterious paracrine-like effects on neighboring CNS cells, contributing to accelerated aging, neurodegeneration (Holloway et al, 2023; Melo Dos Santos et al, 2024), and

the development of various neurodegeneration disorders including Alzheimer's disease (AD) (Dehkordi et al, 2021), Parkinson's disease (PD) (Chinta et al, 2018), amyotrophic lateral sclerosis (Vazquez-Villasenor et al, 2020), and HIV-associated neurocognitive disorders (HAND) (Thangaraj et al, 2021).

Endolysosomes, referring to the endosomal–lysosomal system consisting of endosomes, lysosomes, and autolysosomes, are critical for the degradation of macromolecules or damaged organelles delivered to lysosomes via endocytosis or autophagy and critical for metabolism and cellular homeostasis (Ballabio & Bonifacino, 2020). Endolysosome dysfunction could lead to abnormal accumulation of undegraded materials (macromolecules and mitochondria) in endolysosomes and endolysosome enlargement (Datta et al, 2021b; Khan et al, 2022), mitochondrial dysfunction (Deus et al, 2020; Stepien et al, 2020; Tintos-Hernandez et al, 2021), impaired clearance of viral factors (Khan et al, 2022; Li et al, 2023), augmented release of their luminal contents via exocytosis (Datta et al, 2019; Kim et al, 2021) that contribute to inflammation (Bordon, 2011; Qian et al, 2017; Yambire et al, 2019; Rawnsley & Diwan, 2020; Toyama-Sorimachi & Kobayashi, 2021), and synaptodendritic impairment (Datta et al, 2021a). As such, dysfunction of endolysosome contributes to the development of neurodegeneration disorders including AD (Van Acker et al, 2019; Hung & Livesey, 2021), PD (Muraleedharan & Vanderperre, 2023), amyotrophic lateral sclerosis (Todd et al, 2023), and HAND (Wendie et al, 2024). Emerging evidence indicates that endolysosome dysfunction is strongly linked to cellular senescence (Gorgoulis et al, 2019; Rovira et al, 2022; Curnock et al, 2023; Tan & Finkel, 2023); profound changes of endolysosome structure and function are found in senescent cells, including endolysosome enlargement, endolysosome de-acidification, endolysosome membrane leakage, accumulation of lipofuscin, and up-regulation of endolysosome enzymes, with senescence-associated $\beta$-galactosidase (SA-$\beta$-gal) being the most widely employed marker of the senescent state (Kurz et al, 2000; Lee et al, 2006). However, it is unclear whether endolysosomal dysfunction represents a driver or consequence of cellular senescence (Tan & Finkel, 2023).

The development of HAND is attributed, at least in part, to CNS persistence of HIV-1 Tat (Johnson et al, 2013; Dickens et al, 2017; Henderson et al, 2019; Ajasin & Eugenin, 2020; Marino et al, 2020), an

Department of Biomedical Sciences, University of North Dakota School of Medicine and Health Sciences, Grand Forks, ND, USA

Correspondence: xuesong.chen@und.edu

essential protein for viral transcription (Kameoka et al, 2002) that is actively secreted from HIV-1–infected cells (Ensoli et al, 1990; Chang et al, 1997; Rayne et al, 2010; Agostini et al, 2017). In the brain, HIV-1 Tat is detected on neurons, astrocytes, and other cells (Liu et al, 2000; Donoso et al, 2022), where Tat could induce direct neurotoxic effects and neuroinflammation. Furthermore, Tat has been shown to induce cellular senescence in CNS cells (Thangaraj et al, 2021; Pillai et al, 2023). Such Tat-induced cellular senescence could contribute to the development of accelerated aging, neuro-inflammation, and neurodegeneration in HAND (Cole et al, 2017; Dickens et al, 2017; Mackiewicz et al, 2019; Zhao et al, 2020; Zhao et al, 2022). As secreted proteins, Tat enters endolysosomes via receptor-mediated endocytosis (Frankel & Pabo, 1988; Mann & Frankel, 1991; Liu et al, 2000; Tyagi et al, 2001; Debaisieux et al, 2012; Gaskill et al, 2017), and we have shown that Tat induces endolysosome damage and dysfunction in neurons and astrocytes (Hui et al, 2012; Chen et al, 2013; Khan et al, 2022). Given that endolysosome dysfunction represents an early step in exogenous Tat-induced cellular senescence, the present studies test the hypothesis that Tat induces cellular senescence via an endolysosome-dependent mechanism in human astrocytes.

In this study, we demonstrate that internalized Tat interacts with an endolysosome-resident arginine sensor SLC38A9 (Wang et al, 2015; Wyant et al, 2017; Savini et al, 2019) via its arginine-rich domain and that such an interaction mediates Tat-induced endolysosome dysfunction and senescence-like phenotype in astrocytes.

# Results

## The arginine-rich basic domain is critical for Tat-induced senescence-like phenotype in human astrocytes

Using the SA-$\beta$-gal assay, the most widely employed marker of the senescent state (Kurz et al, 2000; Lee et al, 2006), we determined the extent to which Tat induces the senescence-like phenotype in human astrocytes. We demonstrated that Tat treatment for 48 h increased the percentage of SA-$\beta$-gal–positive cells in a concentration-dependent manner (Fig 1A), a finding that is consistent with other's findings that Tat induces cellular senescence (Thangaraj et al, 2021; Pillai et al, 2023). At the concentrations used, Tat did not induce cytotoxicity as indicated by released LDH activity (Fig 1B). Although Tat concentrations in brain parenchyma are unknown, nanomolar concentrations of Tat have been detected in CSF of HIV-infected individuals on ART drugs (Johnson et al, 2013; Henderson et al, 2019); thus, local concentrations of Tat in brain parenchyma could be quite high. In the present study, we did observe that Tat at a lower concentration (10 nM) significantly increased the percentage of SA-$\beta$-gal–positive cells. We also conducted time-dependent studies of Tat-induced cellular se-nescence. We found that Tat (100 nM) significantly increased the release of IL-6 (an important SASP factor) at 48-h post-treatment, but not at earlier time points (Fig 1C). Furthermore, we assessed the extent to which Tat (100 nM) induces cellular senescence at longer time points, and we found that Tat also significantly increased the percentage of SA-$\beta$-gal–positive cells at 72-h post-treatment, but to

a lesser extent than that at 48-h post-treatment (Fig 1D). Thus, Tat at the concentration of 100 nM for 48 h induces the most robust phenotypes of cellular senescence. With such a robust effect, we can confidently assess whether Tat-induced cellular senescence could be attenuated. Thus, Tat (100 nM) treatment for 48 h was used for subsequent studies.

The arginine-rich basic domain of Tat (amino acids 49–57) has been shown to play an important role in multiple aspects of Tat biology including transactivating activity, nucleolar localization (Endo et al, 1989), neurotoxic effect (Sabatier et al, 1991; Weeks et al, 1995; Hui et al, 2012), and neuroinflammation (Philippon et al, 1994; Ruiz et al, 2019). Here, we explored the extent to which the arginine-rich basic domain affects the Tat-induced senescence-like phe-notype in human astrocytes. Three sets of cellular senescence markers were used (Hernandez-Segura et al, 2018; Gonzalez-Gualda et al, 2021) including protein markers for cell cycle arrest (p16$^{Ink4a}$ and p21$^{CIP1}$), SA-$\beta$-gal activity, and SASP factors (IL-6, IL-8, and CCL2/MCP1). We demonstrated that Tat (100 nM for 48 h), but not mutant Tat (mTat, 100 nM for 48 h), with deletion of the arginine-rich basic domain encompassing amino acids 49–57, significantly increased the percentage of SA-$\beta$-gal–positive cells (Fig 1E), SA-$\beta$-gal activity (Fig 1F), protein levels of p16 (Fig 1G), protein levels of p21 (Fig 1H), and the release of SASP including IL-6 (Fig 1I), IL-8 (Fig 1J), and CCL2 (Fig 1K). Our findings suggest that the arginine-rich domain is critical for the Tat-induced senescence-like phenotype in human astrocytes.

## The arginine-rich basic domain is critical for Tat-induced endolysosome dysfunction

As a secreted viral protein, Tat is known to enter cells through receptor-mediated endocytosis with the assistance of various surface receptors including CD26 (Gutheil et al, 1994; Ohtsuki et al, 2000), CXC chemokine receptor type 4 (Secchiero et al, 1999; Ghezzi et al, 2000; Xiao et al, 2000), heparan sulfate proteoglycans (Ishihara, 2001), low-density lipoprotein receptor–related protein 1 (LRP1) (Liu et al, 2000; Cafaro et al, 2024), vascular endothelial growth factor receptor (Mitola et al, 1997; Arese et al, 2001; Nyagol et al, 2008; Urbinati et al, 2012), and integrin (Urbinati et al, 2005a, 2005b; Monini et al, 2012; Cafaro et al, 2020). We have shown that Tat induces endolysosome dysfunction in neurons and astrocytes (Hui et al, 2012; Chen et al, 2013; Khan et al, 2022). Here, we explore the role of the arginine-rich basic domain in Tat internalization and Tat-induced endolysosome damage. We demonstrated that FITC-labeled Tat (FITC-Tat) entered endolysosomes identified with LysoTracker in human astrocytes at 1 h post-treatment (Fig 2A). Mutant Tat, which lacks the arginine-rich basic domain, is still able to interact with several cell surface receptors, such as CD26, LRP1, CXCR4, and integrins, all of which may facilitate its endocytosis. To assess the extent of mutant Tat entry into the endolysosomes of human astrocytes, we used Alexa 488–labeled mutant Tat (mTat-Alexa 488). We found a substantial intracellular presence of mTat-Alexa 488, which colocalized with endolysosomes marked by LysoTracker at 1 h post-treatment (Fig 2B). This indicates that the arginine-rich domain is not necessary for Tat internalization in human astrocytes.

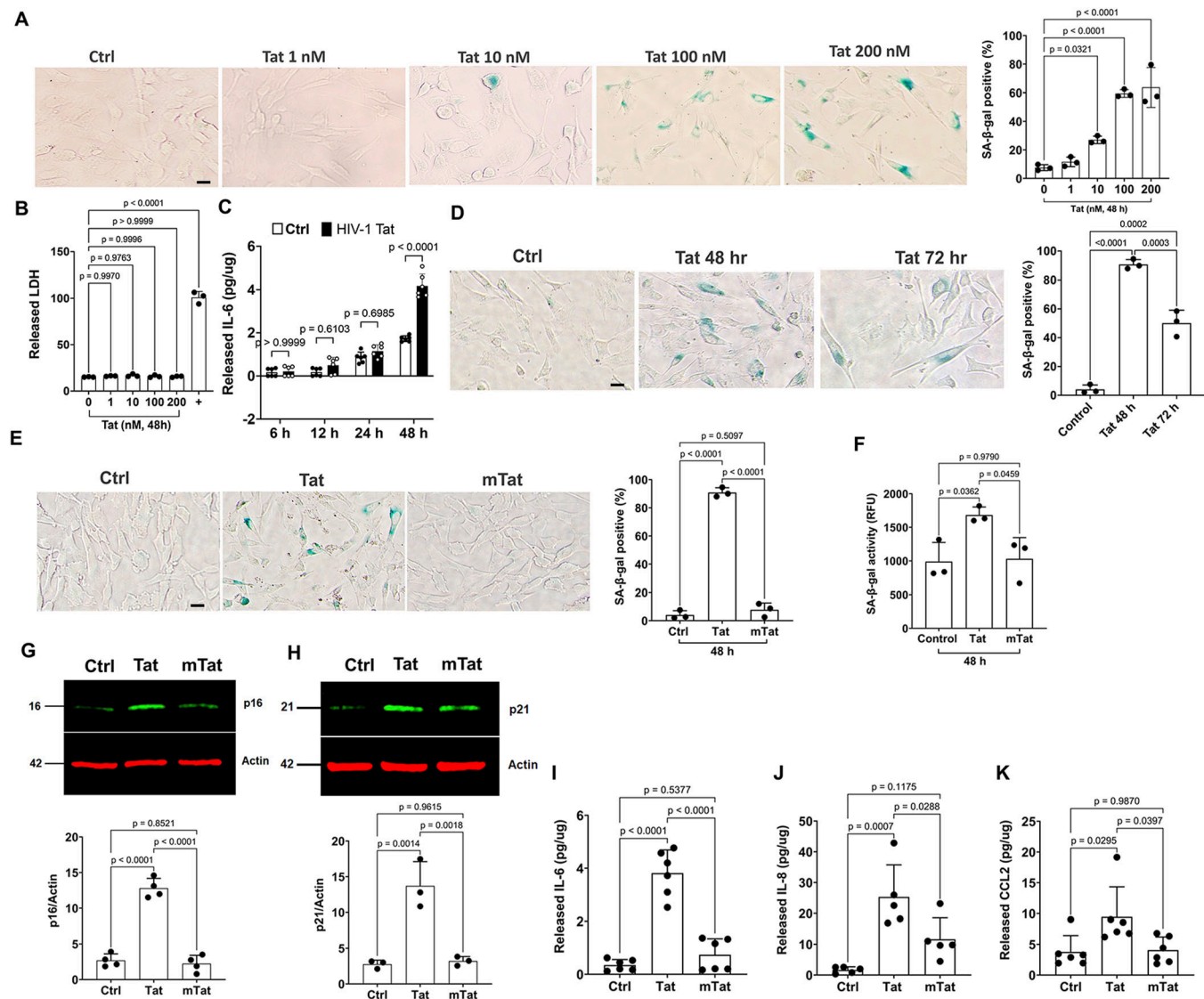

**Figure 1. Arginine-rich domain is critical for the Tat-induced senescence-like phenotype in human astrocytes.**
**(A)** Tat treatment for 48 h increased the percentage of SA-$\beta$-gal–positive cells in a concentration-dependent manner in human astrocytes (n = 3, scale bar = 40 $\mu$m).
**(B)** Tat treatment for 48 h did not increase the release of LDH into the media of cultured astrocytes (n = 3). **(C)** Tat (100 nM) significantly increased the release of IL-6 at 48 h post-treatment (n = 6). **(D)** Tat (100 nM) significantly increased SA-$\beta$-gal–positive cells at 48 and 72 h post-treatment (n = 3, scale bar = 40 $\mu$m). **(E, F)** Tat (100 nM for 48 h), but not mutant Tat (100 nM for 48 h), significantly increased SA-$\beta$-gal–positive cells ((E), n = 3, scale bar = 40 $\mu$m) and elevated SA-$\beta$-gal activity ((F), n = 3) in human astrocytes. **(G, H)** Tat (100 nM, 48 h), but not mutant Tat, significantly increased protein levels of the senescence marker p16Ink4a ((G), n = 4) and p21CIP1 ((H), n = 3) in human astrocytes. **(I, J, K)** Tat (100 nM, 48 h), but not mutant Tat, increased the release of IL-6 ((I), n = 6), IL-8 ((J), n = 5), and CCL2 ((K), n = 6) in the astrocyte culture media. Data information: Data were expressed as means ± SD. n = independent culture preparations. **(C)** Two-way ANOVA followed by Tukey's post hoc test in (C) and one-way ANOVA followed by Tukey's post hoc test for the rest of data.

Next, we explored the role of arginine-rich domain in Tat-induced changes in endolysosome function by measuring endolysosome pH in human astrocytes with a ratiometric method, in which pHLys Green reduces fluorescence as pH increases, whereas LysoPrime Red is pH-resistant. Consistent with our previous findings (Hui et al, 2012; Khan et al, 2022), we demonstrated that Tat (100 nM, 48 h) induced endolysosome de-acidification (Fig 2C), as indicated by the decreased fluorescent ratio of pHLys Green to LysoPrime Red. However, mutant Tat, lacking the arginine-rich basic domain, failed to induce endolysosome de-acidification (Fig 2C).

Our previous studies indicate that Tat induces signs of endolysosome membrane leakage (Hui et al, 2012; Khan et al, 2022), which could lead to the endolysosome de-acidification effect. Thus, we determined the extent to which Tat induces endolysosome membrane leakage using the galectin-3 punctate assay in human astrocytes. As a cytosolic protein, galectin-3 enters the lumen of endolysosomes and forms galectin-3 puncta, when the integrity of the endolysosome membrane is compromised (Aits et al, 2015; Eriksson et al, 2020). We demonstrated that Tat (100 nM, 2 and 24 h) induced endolysosome membrane leakage (Fig

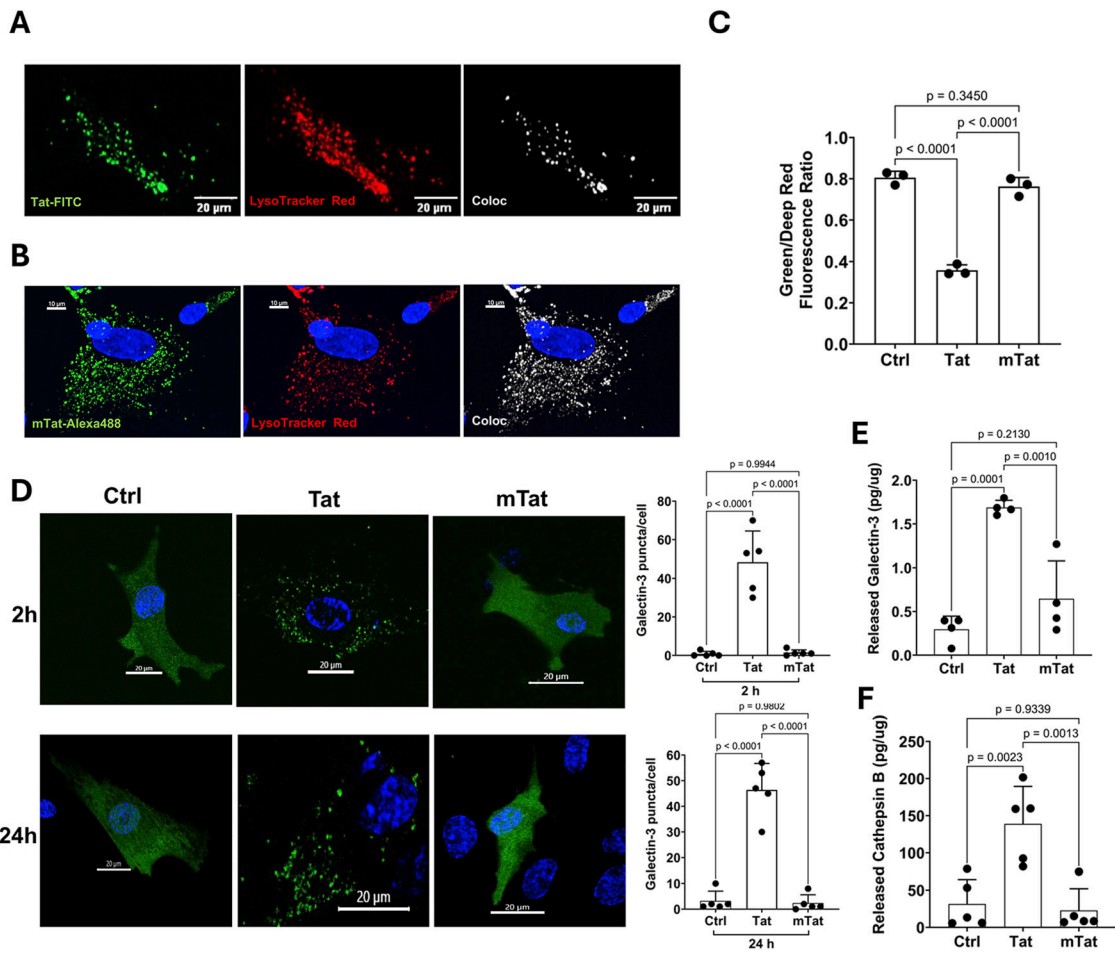

**Figure 2. Arginine-rich domain is critical for Tat-induced endolysosomal dysfunction in human astrocytes.**
**(A)** FITC-labeled Tat (Tat-FITC) colocalized with endolysosomes marked by LysoTracker (red) in human astrocytes at 1 h post-treatment, scale bar = 20 $\mu$m. **(B)** Mutant Tat labeled with Alexa 488 (mTat-Alexa 488) also colocalized with endolysosomes marked by LysoTracker (red) in astrocytes at 1 h post-treatment. Nuclear staining was performed using NucBlue (scale bar = 10 $\mu$m). **(C)** Tat (100 nM for 48 h), but not mutant Tat, led to endolysosome de-acidification, evident from the reduced Green/Deep Red fluorescence ratio (n = 3). **(D)** In astrocytes expressing EGFP-tagged galectin-3, Tat (100 nM), but not mutant Tat, significantly increased galectin-3 punctate formation after 2 and 24 h of treatment (n = 5). NucBlue was used for nuclear staining, scale bar = 20 $\mu$m. **(E, F)** Tat (100 nM for 48 h), but not mutant Tat, elevated galectin-3 levels ((E), n = 4) and cathepsin B levels ((F), n = 5) in the astrocyte culture media. Data information: Data were expressed as means ± SD. n = independent culture preparations. **(C, D, E, F)** One-way ANOVA followed by Tukey's post hoc test in (C, D, E, F).

2D) as indicated by the formation of GFP-galectin-3 puncta. However, mutant Tat, lacking the arginine-rich basic domain, failed to induce endolysosome membrane leakage (Fig 2D). Endolysosome dysfunction can lead to augmented release of their luminal contents via exocytosis (Datta et al, 2019; Kim et al, 2021), and in such a process, a variety of factors can be secreted, including cathepsin B (Verderio et al, 2012; Fan & He, 2016), ATP (Zhang et al, 2007), exosomes (You et al, 2020), and galectin-3 (Popa et al, 2018; Jia et al, 2020). Thus, we determined the extent to which Tat induces the secretion of endolysosome luminal content into media of cultured human astrocytes using the ELISA method. We demonstrated that Tat (100 nM for 48 h), but not mutant Tat lacking the arginine-rich domain, induced the secretion of galectin-3 (Fig 2E) and cathepsin B (Fig 2F) into the extracellular space. Our findings suggest that the arginine-rich domain is critical for Tat-induced endolysosome damage and dysfunction.

### Tat interacts with the endolysosome-resident arginine sensor SLC38A9

Belonging to the solute carrier (SLC) family, SLC38A9 is an amino acid transporter (Schioth et al, 2013; Rebsamen et al, 2015), competent for transporting glutamine, leucine, and arginine. SLC38A9 is an endolysosome membrane protein, with 11 transmembrane helices (Lei et al, 2018). SLC38A9 interacts with the Rag–Regulator complex to activate mammalian target of rapamycin complex 1 (mTORC1) (Schioth et al, 2013), and it has been shown that SLC38A9 signals arginine sufficiency in the lumen of endolysosomes, which is critical for regulating the activity of mTORC1 (Wang et al, 2015; Jung et al, 2015; Rebsamen et al, 2015; Lei et al, 2021). Thus, SLC38A9 encompasses the functions of both a transporter and a receptor, but the signaling may not involve amino acid transport (Lei et al, 2021). Given that SLC38A9 functions as an endolysosome arginine sensor that interacts with v-ATPase (Wang et al, 2015; Wyant et al,

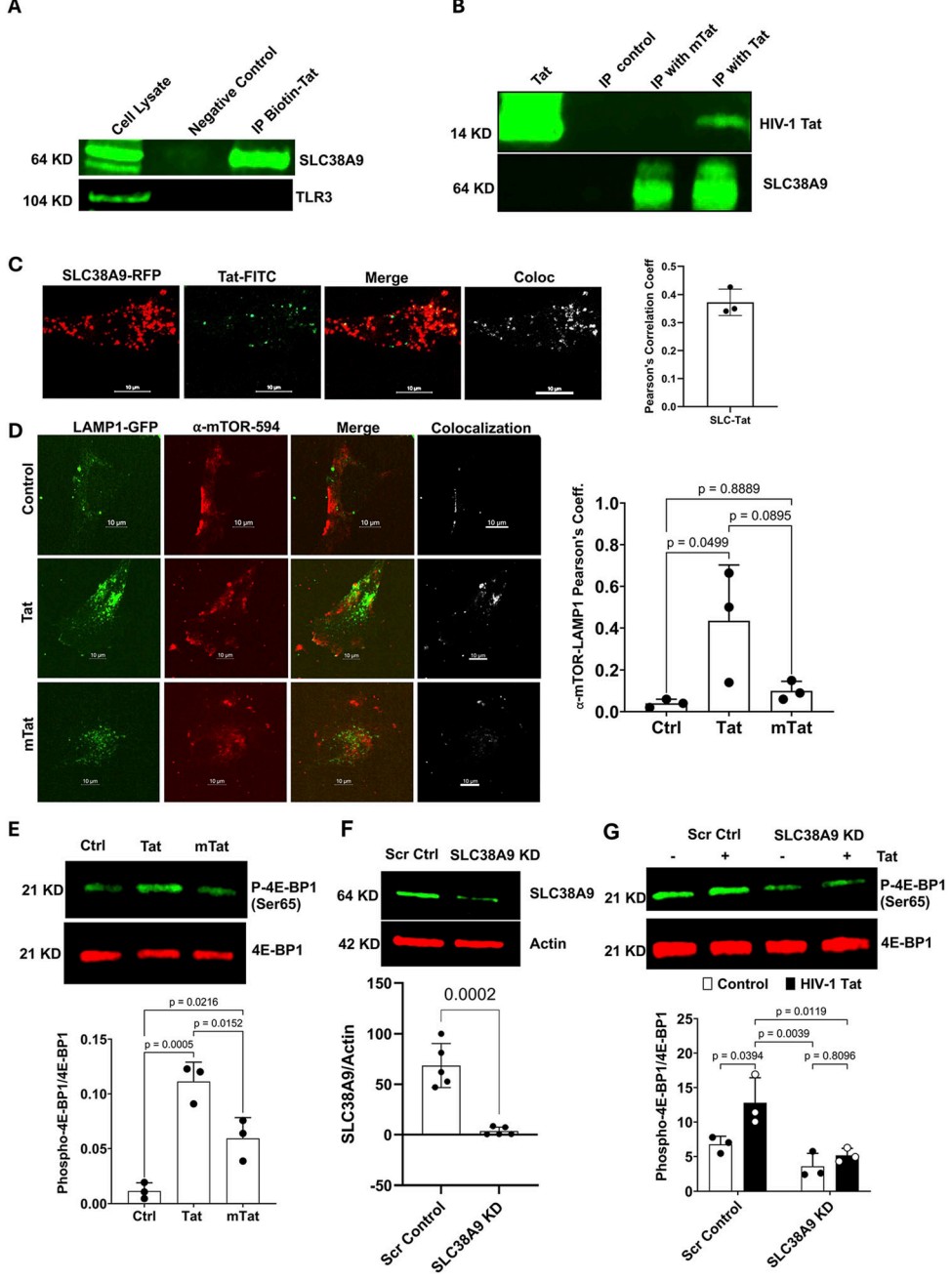

**Figure 3. Tat interacts with the endolysosome-resident arginine sensor SLC38A9.**

**(A)** Biotin-labeled Tat was used as bait to pull down SLC38A9, but not TLR3, from U87MG cell lysates. **(B)** Biotinylated anti-SLC38A9 antibody was used to capture SLC38A9 from U87MG cell lysates, followed by incubation with Tat or mutant Tat. Tat, but not mutant Tat, was detected in the precipitates, whereas a biotinylated isotype IgG served as a negative control. **(C)** SLC38A9-RFP colocalized with Tat-FITC (green), yielding Pearson's correlation coefficient of 0.37 (n = 3, scale bar = 10 $\mu$m). **(D)** Tat (100 nM, 2 h), but not mutant Tat, increased colocalization of $\alpha$-mTOR (red) with LAMP1-GFP in human astrocytes (n = 3, scale bar = 10 $\mu$m). **(E)** Tat (100 nM, 1.5 h) significantly increased phosphorylation of the mTORC1 downstream target 4E-BP1 at serine 65. Mutant Tat (100 nM, 1.5 h) also increased phosphorylation of 4E-BP1, but the extent is lower than that of Tat (n = 3). **(F)** Quantitative immunoblotting confirmed the knockdown of SLC38A9 in human astrocytes using specific siRNAs (n = 5). **(G)** SLC38A9 knockdown attenuated phospho-4E-BP1 and blocked Tat (100 nM, 1.5 h)-induced increases in phospho-4E-BP1 (n = 3). Data information: Data were expressed as means ± SD. n = independent culture preparations. **(D, E, F, G)** Two-tailed $t$ test in (F), one-way ANOVA followed by Tukey's post hoc test in (D, E), and two-way ANOVA followed by Tukey's post hoc test in (G).

2017; Savini et al, 2019), we hypothesize that Tat interacts with SLC38A9 and that such an interaction mediates Tat-induced endolysosome dysfunction and senescence-like phenotype. In a proof-of-concept study, we determined the interactions between Tat and SLC38A9 using the pull-down/immunoprecipitation method. Using biotin-labeled Tat (Fig 3A) as bait proteins, we pulled down SLC38A9 from the U87MG cell lysate, and as a control, TLR3, an endolysosome-resident RNA sensor protein, was not pulled down. In an immunoprecipitation assay, using SLC38A9 antibodies as bait proteins that pull down SLC38A9 from the cell lysate, we also demonstrated that SLC38A9 interacted with Tat (Fig

3B), but not with mutant Tat lacking the arginine-rich domain (Fig 3B). Furthermore, to confirm that Tat could interact with SLC38A9 within endolysosomes in living cells, we performed colocalization studies using live-cell imaging. We demonstrated that FITC-labeled Tat colocalized with SLC38A9-RFP that was transiently expressed in human astrocytes (Fig 3C).

Given that SLC38A9 is an activator of mTORC1 and that activation of mTORC1 occurs at the surface of endolysosomes (Napolitano et al, 2022), where activated mTORC1 is recruited to endolysosome surface (Mutvei et al, 2020), we explore the extent to which Tat induces mTORC1 recruitment to endolysosomes in live human

astrocytes using colocalization studies. Because Tat enters endo-lysosomes 1 h post-treatment, which represents an early step in Tat-induced cellular response, we determined the effect of Tat treatment for up to 2 h on mTORC1 activation. We demonstrate that Tat (100 nM, 2 h) increased the colocalization of α-mTOR-594 with LAMP1-GFP in human astrocytes (Fig 3D). In contrast, the mutant Tat lacking the arginine-rich domain did not increase the colocalization of α-mTOR-594 with LAMP1-GFP (Fig 3D). To further explore the extent to which Tat affects mTORC1 activity, we determine the phosphorylation status of 4E-BP1, a downstream target of mTORC1. We demonstrated that Tat (100 nM, 1.5 h) enhanced mTORC1 activity in human astrocytes, as evidenced by increased phosphorylation of 4E-BP1 at serine 65 (Fig 3E). Although mutant Tat lacking the arginine-rich domain also increased the phosphorylation of 4E-BP1, the extent of 4E-BP1 phosphorylation induced by Tat was significantly greater than that of mutant Tat. To assess the role of SLC38A9 in Tat-induced mTORC1 activation, we knocked down SLC38A9 using the siRNA approach in human astrocytes (Fig 3F). We demonstrated that SLC38A9 knockdown significantly reduced the phosphorylation of 4E-BP1 at serine 65 and blocked Tat-induced phosphorylation of 4E-BP1 (Fig 3G), which suggests that the interaction between Tat and SLC38A9 activates mTORC1 in human astrocytes.

### SLC38A9 knockdown attenuates Tat-induced endolysosome dysfunction, LTR transactivation, and cellular senescence

Next, we determined the extent to which SLC38A9 knockdown affects Tat-induced endolysosome dysfunction in human astrocytes, as indicated by the release of endolysosome contents (galectin-3 and cathepsin B) and the formation of endogenous galectin-3 puncta. We demonstrated that SLC38A9 knockdown significantly attenuated Tat-induced release of galectin-3 (Fig 4A) and cathepsin B (Fig 4B) into the media of human astrocytes. Furthermore, SLC38A9 knockdown significantly attenuated Tat-induced endolysosome membrane leakage, as indicated by the formation of galectin-3 puncta in LAMP-1–positive vesicles, at 24 h post-treatment (Fig 4C). Thus, internalized Tat could interact with SLC38A9 to induce endolysosome dysfunction.

After its internalization into endolysosomes, exogenous Tat must escape from endolysosomes into the cytosol and make its way to the nucleus, where it activates the HIV-1 LTR promoter for viral replication (Ensoli et al, 1993; Vives, 2003; Vendeville et al, 2004). In a published study (Khan et al, 2022), we have demonstrated that extracellular Tat enters astrocytes via endocytosis, that Tat accumulated in endolysosomes is functionally intact, and that upon release from endolysosomes, Tat induces HIV-1 LTR transactivation. Such a process could reactivate latent HIV-1 reservoirs and play an important role in latent infection of HIV-1. Consistent with this notion, it has been shown that autophagy restricts HIV-1 infection by selectively degrading Tat in CD4[+] T lymphocytes (Sagnier et al, 2015).

Thus, the process whereby Tat induces endolysosome dysfunction could impair the capability of endolysosomes to degrade internalized Tat. Because SLC38A9 knockdown attenuates Tat-induced endolysosome dysfunction, we have assessed the extent to which SLC38A9 knockdown affects cellular levels of Tat in human astrocytes treated with Tat (2 μg/ml) for 48 h. We found that

SLC38A9 knockdown decreased cellular levels of Tat (Fig 4D), indicating that SLC38A9 knockdown enhances Tat degradation and increases the bioavailability of Tat. Furthermore, the process whereby Tat induces endolysosome membrane leakage via its interaction with SLC38A9 could facilitate the escape of Tat from endolysosomes into the cytosol and subsequent activation of HIV-1 LTR in the nucleus. Using U87MG cells that stably express HIV-1 LTR with a luciferase reporter gene, we determined the extent to which SLC38A9 knockdown affects Tat-mediated HIV-1 LTR transactivation. We demonstrated that shRNA knockdown of SLC38A9 (Fig 4E) significantly attenuated extracellular Tat-mediated HIV-1 LTR transactivation (Fig 4F). Thus, our findings suggest that the interaction between Tat and endolysosome-resident SLC38A9 could reactivate latent HIV-1 reservoirs and thus play an important role in latent infection of HIV-1.

To assess the role of SLC38A9 in Tat-induced cellular senescence, we investigated the effect of siRNA knockdown of SLC38A9 on the Tat-induced senescence-like phenotype in human astrocytes as indicated by increased release of SASP, enhanced SA-β-gal activity, and increased p16 levels. We demonstrated that SLC38A9 knockdown significantly attenuated Tat-induced increased release of IL-6 (Fig 4G), IL-8 (Fig 4H), and CCL2 (Fig 4I). SLC38A9 knockdown also significantly attenuated Tat-induced enhanced activity of SA-β-gal (Fig 4J) and increased protein levels of p16[Ink4a] (Fig 4K). These findings indicate that the interaction between Tat and the endolysosome-resident protein SLC38A9 plays a pivotal role in driving Tat-induced cellular senescence in human astrocytes.

## Discussion

Prominent findings of the present study are that internalized Tat interacts with an endolysosome-resident arginine sensor SLC38A9 via the arginine-rich basic domain and that such an interaction leads to endolysosome dysfunction, cellular senescence, and enhanced HIV-1 LTR transactivation.

The combined antiretroviral therapy has successfully suppressed HIV-1 and dramatically increased the life expectancy in people with HIV (PWH) (Katz & Maughan-Brown, 2017; Trickey et al, 2023). However, antiretroviral therapy does not eliminate HIV-1, and reservoirs of HIV-1 in the periphery (Eisele & Siliciano, 2012) and in the brain (Marban et al, 2016; Lutgen et al, 2020) persist in PWH. Furthermore, as PWH live longer, they face a variety of age-related comorbidities including HAND, the prevalence of which remains high (30–50%) even in the combined antiretroviral therapy era (Saylor et al, 2016; Wang et al, 2020; Zenebe et al, 2022). Although the underlying pathogenesis remains elusive, accelerated aging and chronic neuroinflammation are critical for the development of neurodegeneration in HAND (Cole et al, 2017; Mackiewicz et al, 2019). Emerging evidence indicates that cellular senescence present in the brain of PWH (Thangaraj et al, 2021) may underlie the accelerated aging, chronic inflammatory state, and neurodegeneration. One of the important HIV-related factors that contribute to accelerated aging and neurodegeneration in HAND (Cole et al, 2017; Dickens et al, 2017; Mackiewicz et al, 2019; Zhao et al, 2020; Zhao et al, 2022) is the persistence of Tat, which is actively secreted

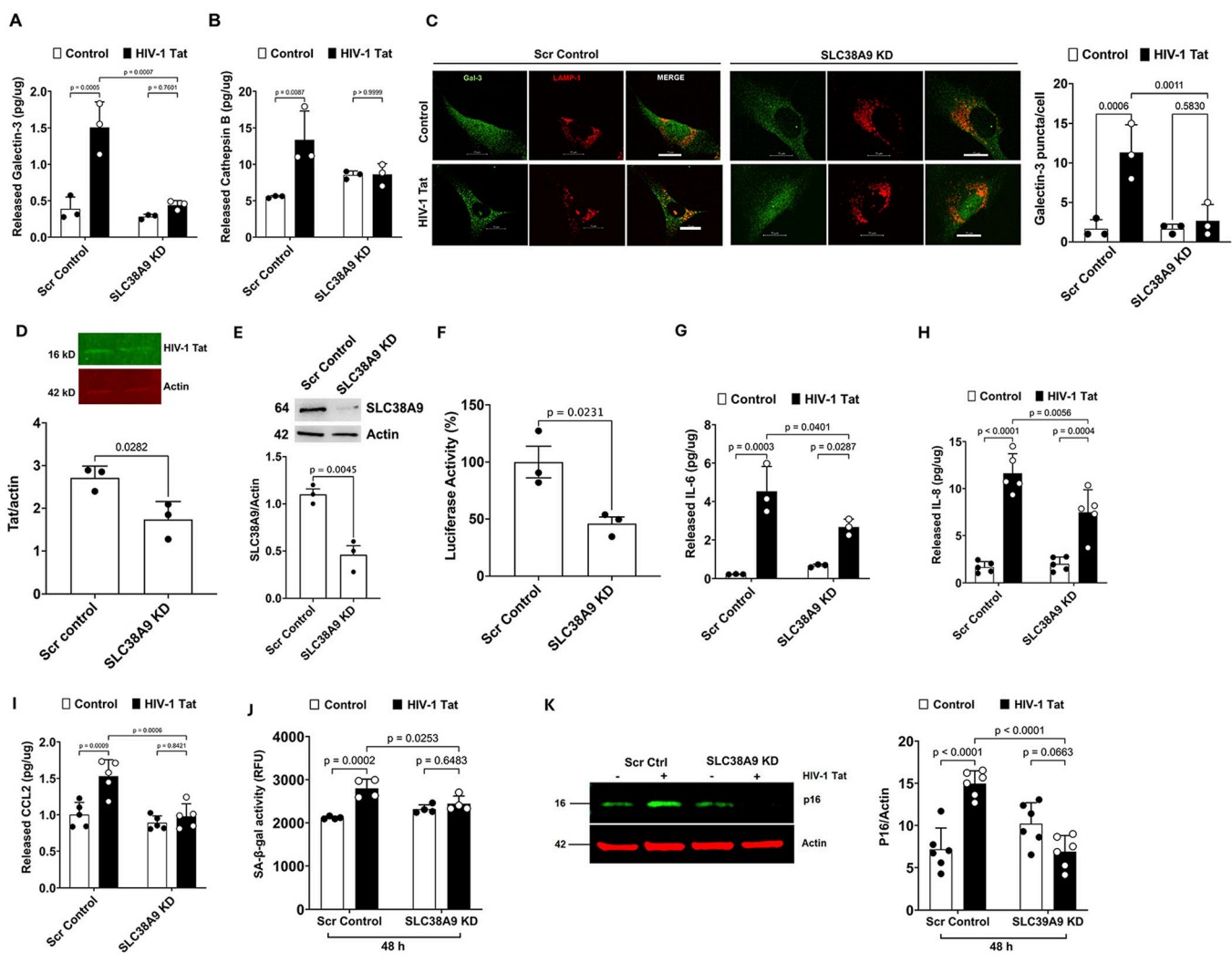

**Figure 4. SLC38A9 knockdown attenuates Tat-induced endolysosome dysfunction, LTR transactivation, and cellular senescence.**
**(A, B)** Knockdown of SLC38A9 alone did not affect the release of endolysosome factors but significantly reduced Tat (100 nM, 48 h)-induced increases in galectin-3 ((A), n = 3) and cathepsin B ((B), n = 3) in the culture media of human astrocytes. **(C)** SLC38A9 knockdown significantly attenuated Tat-induced endolysosome membrane leakage, as indicated by the formation of endogenous galectin-3 puncta in LAMP-1–positive vesicles, at 24 h post-treatment (n = 3), scale = 15 $\mu$m. **(D)** SLC38A9 knockdown significantly decreased the cellular level of Tat at 48 h post-treatment (n = 3). **(E)** Quantitative immunoblotting confirmed the knockdown of SLC38A9 in U87MG cells using specific shRNAs (n = 3). **(F)** Knockdown of SLC38A9 reduced Tat-mediated HIV-1 LTR transactivation in U87MG cells stably transfected with HIV-1 LTR-luciferase reporter (n = 3). **(G, H, I)** SLC38A9 knockdown alone did not affect the release of inflammatory factors; however, SLC38A9 knockdown significantly attenuated Tat (100 nM, 48 h)-induced increases in IL-6 ((G), n = 3), IL-8 ((H), n = 5), and CCL2 ((I), n = 5) in media of cultured human astrocytes. **(J)** SLC38A9 knockdown significantly attenuated Tat (100 nM, 48 h)-induced increases in SA-$\beta$-gal activity in human astrocytes (n = 4). **(K)** SLC38A9 knockdown significantly attenuated Tat (100 nM, 48 h)-induced increases in p16$^{Ink4a}$ protein levels in human astrocytes (n = 6). Data information: Data were expressed as means ± SD. n = independent culture preparations. Two-tailed t test in (D, E, F) and two-way ANOVA followed by Tukey's post hoc test in the rest of data.

from HIV-1–infected cells (Ensoli et al, 1990; Chang et al, 1997; Rayne et al, 2010; Agostini et al, 2017). Nanomolar concentrations of Tat have been detected in the sera (Xiao et al, 2000) and CSF (Henderson et al, 2019) of the PWH. Significantly, current anti-HIV strategies do not block the secretion of Tat (Mediouni et al, 2012), and brain levels of Tat remain elevated, even when HIV-1 levels are below detectable levels (Johnson et al, 2013; Henderson et al, 2019). Furthermore, Tat has been shown to induce cellular senescence in CNS cells (Thangaraj et al, 2021; Pillai et al, 2023).

Because Tat is present on astrocytes (Liu et al, 2000) and plays an important role in astrocyte-mediated neuroinflammation (Conant

et al, 1998; Nath et al, 1999; Kutsch et al, 2000; El-Hage et al, 2005; Blanco et al, 2008; Williams et al, 2009; Henderson et al, 2012; Tewari et al, 2015; Priyanka et al, 2020), and because astrocyte dysfunction and the development of cellular senescence not only result in the loss of their physiological support to neurons, but also result in the release of SASP that elicits deleterious paracrine-like effects on neighboring cells such as neurons, contributing to cognitive impairment (Csipo et al, 2020; Meldolesi, 2023), the present study determined the extent to which Tat induces cellular senescence in human astrocytes. We demonstrated that exogenously added Tat induces the robust senescence-like phenotype in

human astrocytes, as induced by increased protein levels of p16 and p21 that could lead to cell cycle arrest (Tchkonia et al, 2013), increased activity of SA-β-gal (a lysosomal hydrolase) the most widely used markers of senescence (Dimri et al, 1995), and increased release of SASP factors including IL-6, IL-8, and CCL2/MCP1.

When exploring the underlying mechanisms whereby Tat induces the senescence-like phenotype in human astrocytes, we focused on involvement of endolysosomes, because endolysosome dysfunction has strongly linked to cellular senescence (Gorgoulis et al, 2019; Rovira et al, 2022; Curnock et al, 2023; Tan & Finkel, 2023), and it has been shown that inhibiting endolysosome degradation exacerbates the phenotypes of senescence (Qi et al, 2024). Silencing transcription factor EB, which functions as a master regulator of the autophagy–lysosome pathway, also exacerbates senescence (Suzuki et al, 2024). This evidence indicates that inhibiting endolysosome function could drive the development of senescence. Furthermore, because cellular uptake of exogenous Tat into endolysosomes (Mann & Frankel, 1991; Liu et al, 2000; Tyagi et al, 2001; Vendeville et al, 2004; Ajasin & Eugenin, 2020) represents the early step whereby exogenous Tat affects other cellular function, and because our published findings demonstrate that Tat accumulates in endolysosomes of astrocytes and induces endolysosome damage (Khan et al, 2022), we encountered a unique system to determine whether Tat-induced endolysosome dysfunction represents a driver or consequence of cellular senescence.

In the present study, we demonstrated that the arginine-rich basic domain is critical for the Tat-induced endolysosome dysfunction and senescence-like phenotype in astrocytes. Besides the neurotoxic arginine-rich basic domain (49–57) (Sabatier et al, 1991; Tyagi et al, 2001; Buscemi et al, 2007; Hui et al, 2012; Ruiz et al, 2019), Tat is composed of several other domains, with distinct functions crucial for viral replication and pathogenesis. Proline-rich domain (1–20) is important for stabilizing Tat's binding to the inner leaf of the cell membrane (Rayne et al, 2010); cysteine-rich domain (21–37) is important for dimerization and metal binding, playing a critical role in the activation of HIV genomic DNA transcription (Wei et al, 1998), and it is also involved in the binding of Tat to TLR4 (Ben Haij et al, 2015) and the CXCR4 (Secchiero et al, 1999; Ghezzi et al, 2000; Xiao et al, 2000); core domain (38–48) is important for binding to the CDK9-associated C-type cyclin, which is crucial for Tat's transactivation activity (Wei et al, 1998), and it is also involved in the interaction of Tat with LRP1 (Liu et al, 2000; Chen et al, 2016; Cafaro et al, 2024); glutamine-rich domain (58–71) plays a role in interacting with the TAR region of HIV RNA, and in addition, it is implicated in Tat-mediated apoptosis (King et al, 2006); RGD domain (72–85) is crucial for its interaction with integrins, which is important for Tat's involvement of cellular adhesion and signaling processes (Brake et al, 1990; Barillari et al, 1993; Urbinati et al, 2005a, 2005b; Monini et al, 2012; Cafaro et al, 2020); and C-terminal domain (86–101) is important for NF-κB activity (Li et al, 2024), a crucial factor in regulating immune responses and inflammation. Although these regions of Tat could induce inflammatory responses and other toxic effects, our findings suggest that the arginine-rich domain is critical for the Tat-induced senescence-like phenotype, because mutant Tat lacking the arginine-rich basic domain failed to induce endolysosome dysfunction and senescence-like phenotype.

The arginine-rich basic domain is also critical for the interaction between Tat and an endolysosome-resident arginine sensor SLC38A9; mutant Tat lacking the arginine-rich domain does not interact robustly with SLC38A9. Furthermore, we demonstrated that the interaction between Tat and SLC38A9 leads to endolysosome dysfunction and cellular senescence; knocking down SLC38A9 attenuates HIV-1 Tat–induced endolysosome dysfunction and senescence-like phenotype. Thus, our findings suggest that endolysosome dysfunction represents a cause of cellular senescence, and we provide the first evidence that SLC38A9 could sense the luminal environment of endolysosomes and contribute to cellular senescence. Although our findings provide evidence that Tat at the endolysosome lumen could interact with SLC38A9 and induce cellular senescence, such findings do not exclude the possibility that released Tat from endolysosome could induce cellular senescence via SLC38A9-independent mechanisms, which warrants further investigation.

Besides sensing arginine content in the lumen of endolysosomes (Wyant et al, 2017), SLC38A9 could also sensor cholesterol in the lumen of endolysosomes (Castellano et al, 2017), and thus, SLC38A9 may play a role in the recent findings that accumulation of cholesterol in lysosomes maintains the SASP (Roh et al, 2023). Although the underlying signaling warrants further investigation, it is likely that interaction between Tat and SLC38A9 induces cellular senescence via the activation of mTORC1, especially when endolysosomes represent the central platform for mTORC1 activation (Ballabio & Bonifacino, 2020) and mTORC1 activation has been shown to drive many senescence-like phenotypes (Herranz et al, 2015; Laberge et al, 2015). Currently, it is not clear whether SLC38A9-mediated mTORC1 activation leads to SASP induction and senescence dependent or independent of endolysosome damage. However, a recent study has shown that mTORC1 activation induces disassembly of v-ATPase on endolysosomes, thus impairing the degradation capability of endolysosomes. Could mTORC1 activation also lead to endolysosome damage? Although this is an open question, the proteolipid ring (c-ring) of the $V_0$ sector of v-ATPase can form a protein core with a diameter of 3.5 nm (Couoh-Cardel et al, 2016), which opens in the presence of calcium (Peters et al, 2001). Thus, it is possible that mTORC1 activation could lead to endolysosome membrane leakage, which could then contribute to cellular senescence (Suzuki et al, 2024). On the other hand, endolysosome membrane leakage could lead to inhibition of mTORC1 (Jia et al, 2018) likely as a protective mechanism. Thus, the question of whether SLC38A9-mediated mTORC1 activation leads to senescence dependent or independent of endolysosome damage warrants further investigation.

Such Tat-induced endolysosome dysfunction and senescence-like phenotype in astrocytes via its interaction with SLC38A9 could play an important role in the pathogenesis of HAND. Because astrocytes are critical for CNS physiology by providing neurotrophic support, facilitating synaptic signaling, and maintaining the blood–brain barrier, in the present study, we only focused on how Tat-SLC38A9 interaction affects astrocyte function (endolysosome function and cellular senescence), and we have not yet explored the consequence of how such astrocyte dysfunction may affect other CNS cells. Because endolysosomes in astrocytes play a critical role in maintaining a healthy nervous system (Kreher et al, 2021),

and endolysosome dysfunction in astrocytes alone leads to neurodegeneration (Di Malta et al, 2012), Tat-induced endolysosome dysfunction in astrocytes could lead to neurodegeneration. Furthermore, the development of cellular senescence in astrocytes not only results in the loss of their physiological support to neurons but also results in the release of SASP that elicits deleterious paracrine-like effects on neighboring cells such as neurons, contributing to cognitive impairment (Csipo et al, 2020; Meldolesi, 2023). Besides its potential role in neuroinflammation and neurodegeneration, Tat-induced endolysosome dysfunction via its interaction with SLC38A9 could also play an important role in latent infection of HIV-1; when entering astrocytes via endocytosis, exogenous HIV-1 Tat could interact with SLC38A9 and induce endolysosome membrane leakage, which enables the release of Tat and its transition to the nucleus to activate the HIV-1 LTR promoter. Because up to 19% of astrocytes carry HIV-1 DNA in HIV-1–infected brain (Trillo-Pazos et al, 2003; Churchill et al, 2009), such Tat-induced HIV-1 LTR transactivation in astrocytes could play an important role in latent infection of HIV-1.

In summary, the interaction between Tat and SLC38A9 plays a critical role in Tat-induced endolysosome dysfunction, cellular senescence, and enhanced HIV-1 LTR transactivation. Our findings suggest that endolysosome dysfunction represents a cause of cellular senescence and that SLC38A9 represents a novel therapeutic target against senescence and the development of HAND and other neurovegetative diseases.

# Materials and Methods

### Cells

Human primary astrocytes (#1800; ScienCell) were grown in cell culture plates coated with poly-L-lysine, using astrocyte medium supplemented with 2% FBS, 1% astrocyte growth supplement, and 1% penicillin–streptomycin, as per the manufacturer's instructions. Human U87MG cells (#HTB-14; ATCC) were stably transduced with a luciferase gene driven by an HIV-1 Tat–dependent LTR promoter and selected using neomycin (Khan et al, 2022). U87MG cells stably transfected with HIV-1 LTR-luciferase reporter were cultured in DMEM supplemented with 10% FCS and 1% penicillin–streptomycin. Both cell types were incubated at 37°C in a humidified atmosphere containing 5% $CO_2$.

### Fluorescent labeling of recombinant mutant Tat

The recombinant mutant HIV-1 Tat Bal, which lacks the arginine-rich basic domain (Cat# 1062; ImmunoDX), was first dialyzed using a Slide-A-Lyzer Mini dialysis device (Cat# 69562; Thermo Fisher Scientific) with a 7 kD molecular weight cutoff membrane to replace Tris with PBS (pH 7), as the amine-containing Tris interferes with labeling. The dialysis was performed overnight at 4°C with several changes of dialysate. Afterward, fluorescent labeling of the recombinant mutant Tat was carried out using the Alexa Fluor 488 microscale protein labeling kit (Cat# A3006; Thermo Fisher Scientific) following the manufacturer's protocol. In brief, mutant Tat (1

mg/ml) was incubated with the dye at a 2:1 M ratio (dye: mutant Tat) for 15 min at room temperature, targeting the amine termini. After labeling, the conjugate was purified using a spin filter to remove any unbound dye.

### LDH cytotoxicity assay

An LDH cytotoxicity assay kit (Cat# C20300; Invitrogen) was used to assess the cytotoxicity of various reagents on human astrocytes. Briefly, human astrocytes were treated with various concentrations of recombinant full-length (1–101) Tat of HIV-1 Bal (Cat# 1052; ImmunoDX), and recombinant mutant Tat of HIV-1 Bal lacking the arginine-rich domain (Cat# 1062; ImmunoDX) for 48 h at 37°C, with a 10X lysis buffer as a positive control. After treatment, the cell culture medium was collected and LDH activity was measured following the provided protocol. The absorbance was measured at 490 and 680 nm using a microplate reader (BioTek). The calculated absorbance by subtracting the 680 nm value from the 490 nm value was used as relative cytotoxicity.

### SLC38A9 knockdown

To knock down SLC38A9 in human astrocytes, cells were transfected with ON-TARGETplus Human SLC38A9 siRNA SMARTpool (50 nM, Dharmacon Reagents, #L-007337-02-0005; Horizon Discovery) and ON-TARGETplus Non-targeting Pool (50 nM, Dharmacon Reagents, #D-001810-10-05; Horizon Discovery) as a control. SLC38A9 siRNA and control siRNA were dissolved in ddH2O. Lipofectamine 2000 (Cat# 11668019; Invitrogen) was used as the transfection reagent. Transfection was conducted for 6 h, followed by Tat treatment. To stably knock down SLC38A9 in U87MG cells, SLC38A9 shRNA lentiviral particles (Cat# sc-91984-V; Santa Cruz) and control shRNA lentiviral particles-A (Cat# sc-108080; Santa Cruz) were used. Knockdown efficiency was evaluated by immunoblotting.

### Live-cell imaging

To assess the internalization of Tat, human astrocytes were incubated with FITC-labeled Tat protein (4 μg/ml, Cat# 1002-F; ImmunoDX) or Alexa Fluor 488–labeled mutant Tat (4 μg/ml), along with LysoTracker Red DND-99 (50 nM, Cat# L7528; Invitrogen), for 1 h at 37°C. In the colocalization study, human astrocytes were transfected with a SLC38A9-RFP plasmid (Cat# PS100049; OriGene). After 48 h, the cells were treated with 100 nM FITC-labeled Tat protein (Cat# 1002-F; ImmunoDX) or 50 nM LysoTracker Red DND-99 (Cat# L7528; Invitrogen) for 1 h at 37°C. The cells were then washed three times with PBS. For the α-mTOR and LAMP1 colocalization study, human astrocytes were transduced with LAMP1-GFP (Cat# C10596; Thermo Fisher Scientific). After 48 h, the cells were treated with HIV-1 Tat (Cat# 1062; ImmunoDX) or mutant Tat (Cat# 1062; ImmunoDX) at 100 nM for 2 h. After the treatment, cells were initially fixed with 4% PFA in PBS for 15 min. After fixation, cells were washed with PBS and permeabilized with 0.3% Triton X-100 in PBS for 10 min. After another PBS wash, the cells were blocked with 3% goat serum for 1 h at room temperature. Primary antibody incubation was performed overnight at 4°C using mTOR antibody (Cat# MA5-31505, dilution 1:200; Invitrogen). After secondary antibody incubation, cells were washed with

PBS-T and PBS (two washes for 5 min each) and incubated for 2 h at room temperature with goat anti-mouse Alexa Fluor 594 secondary antibody (Cat# ab150116, dilution 1:500; Abcam). After additional PBS-T and PBS washes, imaging was conducted using a Zeiss LSM 800 confocal microscope. The images were subsequently analyzed using Imaris 10.1 software.

## Immunoblotting

Human astrocytes were lysed using 1× RIPA lysis buffer (Cat# 89900; Thermo Fisher Scientific) with a 1× protease inhibitor cocktail (Cat# 78441; Thermo Fisher Scientific). The lysates were centrifuged at 12,000$g$ for 20 min at 4°C, and the supernatants were collected. Protein concentrations were measured using the Bradford protein assay (Bio-Rad). Proteins (20 $\mu$g) were resolved by SDS–PAGE on a 4–12% gel and transferred to PVDF membranes with the iBlot 3 dry transfer system (Invitrogen). The membranes were incubated overnight at 4°C with primary antibodies, with actin antibodies serving as controls (Cat# ab179467, dilution 1:3,000; Abcam and/or Cat# NBP1-47423, dilution 1:3,000; Novus). The primary antibodies used included HIV-1 Tat (Cat# 1302, dilution 1:1,000; ImmunoDX and/or Cat# sc-65913, dilution 1:250; Santa Cruz), SLC38A9 (Cat# PA5-98670, dilution 1:250; Thermo Fisher Scientific), TLR3 (Cat# PA5-20183, dilution 1:1,000; Thermo Fisher Scientific), LRP1 (Cat# ab92544, dilution 1:500; Abcam), p16-INK4A (Cat# 10883-1-AP, dilution 1:500; Proteintech), p21 (Cat# 2947S, dilution 1:400; Cell Signaling), p-4E-BP1 (62.Ser 65) (Cat# sc-293124, dilution 1:150; Santa Cruz), eIF4EBP1 (Cat# ab32024, dilution 1:500; Abcam). After primary antibody incubation, the membranes were treated with fluorescently conjugated secondary antibodies, including goat anti-mouse IgG (Cat# 926-68070, 926-32210, dilution 1:5,000; LI-COR) and goat anti-rabbit IgG (Cat# 926-32211, 926-68071, dilution 1:5,000; LI-COR). Protein band density was quantified using Li-COR Odyssey Fc Imaging System (Li-COR).

## Immunoprecipitation

A pull-down assay was performed using EZ-Link Desthiobiotinylation and Pull-Down Kit (Cat# 16138; Thermo Fisher Scientific). In this process, 10 $\mu$g of biotinylated Tat HIV-1 IIIB (Cat# 1002-B; ImmunoDX) was combined with 50 $\mu$l of streptavidin agarose resin and incubated for 30 min at room temperature. Non-biotinylated HIV-1 Tat was used as a negative control. U87MG cells were lysed in NP-40 lysis buffer (Cat# J60766-AK; Thermo Fisher Scientific) containing a 1× protease inhibitor cocktail (Cat# 78441; Thermo Fisher Scientific). After centrifugation at 12,000$g$ for 20 min at 4°C, the supernatants were collected. The cell lysates (400 $\mu$l) were pre-cleared with streptavidin-conjugated resins and incubated with the resin containing biotinylated bait protein overnight at 4°C. After washing, co-immunoprecipitants were eluted with the provided elution buffer (Cat# 16138; Thermo Fisher Scientific). The eluted samples were subjected to SDS–PAGE and immunoblotting to detect target proteins, including SLC38A9, TLR3, TLR4, TLR8, and TLR9. U87MG cell lysate was used as a positive control.

In a separate experiment using Biotinylated Protein Interaction Pull-Down Kit (Cat# 21115; Thermo Fisher Scientific), 400 $\mu$l of cell lysates was incubated with 30 $\mu$g of biotinylated anti-SLC38A9

antibody (Cat# LS-C679509-50; LS Bio) overnight at 4°C. A biotin-rabbit anti-mouse IgG secondary antibody (Cat# SA5-10238; Thermo Fisher Scientific) was used as an isotype IgG control. The mixture was then incubated with either HIV-1 Tat or HIV-1 mutant Tat (10 $\mu$g protein in 90 $\mu$l TBS) overnight at 4°C. After washing, co-immunoprecipitants were eluted with the provided elution buffer (Cat# 21115; Thermo Fisher Scientific). The eluted samples were subjected to SDS–PAGE (4–12%) and immunoblotting to detect HIV-1 Tat proteins.

## ELISA

The release of inflammatory factors from human astrocytes was quantified using several ELISA kits: Human IL-6 ELISA kit (Cat# ab100572; Abcam), Human IL-8 ELISA kit (Cat# ab46032; Abcam), Proteome Profiler Human Cytokine Array kit (Cat# ARY005B; R&D Systems), Human MCP1 ELISA kit (Cat# ab100586; Abcam), Human Cathepsin B ELISA kit (Cat# ab119584; Abcam), Human Galectin-3 ELISA kit (Cat# ab269555; Abcam), Human IL-18 ELISA kit (Cat# BMS267-2; Thermo Fisher Scientific), Human IFN-gamma ELISA kit (Cat# KHC4021; Thermo Fisher Scientific), Human IL-1 alpha ELISA kit (, Cat# BMS243-2; Thermo Fisher Scientific), Human IFN-alpha ELISA kit (Cat# BMS216; Thermo Fisher Scientific), Human IFN-beta ELISA kit (Cat# QK410; R&D Systems), Human TNF-alpha ELISA kit (Cat# BMS223-4; Thermo Fisher Scientific), Human IL-12 ELISA kit (Cat# ab46035; Abcam), and Human Complement C3 ELISA kit (Cat# ab108823; Abcam).

After treatment, the cell culture supernatants were collected and centrifuged at 1,500$g$ for 2 min to remove any cellular debris. According to the manufacturer's protocols, the supernatants (in triplicates) or standards (in duplicates) were added to the pre-coated wells and incubated overnight at 4°C. After washing, biotinylated detection antibodies were added to the wells, followed by incubation with HRP-conjugated streptavidin. TMB substrate was then added, allowing color development for 30 min. The reaction was halted by adding a stop solution, and absorbance was measured at 450 nm using a BioTek microplate reader. The concentrations of inflammatory factors were calculated using standard curves with known concentrations of specific inflammatory factors, employing a four-parameter logistic curve fitting in Gen5 software (BioTek Instruments, Inc.). These concentrations were normalized to the total protein content of the cultured cells, which was determined using the Bradford protein assay.

## SA-$\beta$-gal activity assay

The SA-$\beta$-gal activity was assessed using an SA-$\beta$-gal activity assay kit (Cat# ENZ-KIT129; Enzo Life Sciences) following the manufacturer's instructions. In brief, cells were lysed with 1X cell lysis buffer and incubated at 4°C for 15 min. The lysates were then centrifuged at 12,000$g$ for 10 min at 4°C. The resulting supernatants, collected in triplicates, were transferred to a 96-well plate. 2X assay buffer was added, and the plate was incubated at 37°C for 3 h, protected from light and without $CO_2$. The reaction was stopped by adding a stop solution, and fluorescence was measured using a microplate reader (BioTek) with an excitation wavelength of 360 nm and an emission

wavelength of 465 nm. The SA-$\beta$-gal activity was expressed as relative fluorescent units, normalized to the total protein content.

## SA-$\beta$-gal staining

The staining of $\beta$-galactosidase was performed using Senescence $\beta$-Galactosidase Staining Kit (#9860; Cell Signaling). In summary, post-treatment cells were fixed with 1X Fixative Solution for 20 min at room temperature. After fixation, the cells were rinsed with PBS, and a $\beta$-galactosidase staining solution, adjusted to pH 6.0, was applied. The cells were then incubated overnight at 37°C in a $CO_2$-free, dry incubator. Post-incubation, the cells were observed under a microscope (Olympus) at 200X magnification to detect the development of a blue color, indicative of $\beta$-galactosidase activity.

## HIV-1 LTR transactivation

U87MG cells stably transfected with HIV-1 LTR-luciferase reporter were plated at a density of 30–40% confluency (~10,000 cells per well) in 96-well plates. The cells were treated with 2 $\mu$g/ml HIV-1 (IIIB) Tat (Cat# 1002; ImmunoDX) after chloroquine treatment (100 $\mu$M for 4 h). After 48 h of incubation, luciferase activity was measured using a Promega luciferase assay system (Cat# E2510; Promega). Relative luminescence units were quantified with a fluorometer/luminometer plate reader (SpectraMAX GEMINI EM; Molecular Devices).

## Galectin-3 punctate assay

Human astrocytes were seeded at a density of 15,000 cells per 35-mm dish and transfected the next day with the pEGFP-hGal3 plasmid (Cat# 73080; Addgene), which expresses EGFP-tagged galectin-3. 48 h after transfection, cells were treated with HIV-1 Tat (100 nM), or HIV-1 mutant Tat (100 nM), for 2 h or 24 h. To assess the formation of endogenous galectin-3 puncta, cells were treated with HIV-1 Tat (100 nM) for 24 h. After treatment, cells were stained with LAMP1 (H4A3; Santa Cruz) and galectin-3 (sc-23938; Abcam). All images were acquired on an Andor DragonFly 200 platform using a cf40 Zyla camera attached to a Leica DMi8 confocal microscope using Fusion software. Images were exported to tiff format and the puncta counted using ImageJ.

## Endolysosome pH measurement

The lysosomal acidic pH detection kit (item code L268-10; Dojindo) was employed to measure the pH of endolysosomes. Human astrocytes were cultured in 35-mm dishes and treated with HIV-1 Tat (100 nM) or mutant Tat (100 nM) for 48 h. After treatment, cells were washed twice with serum-free medium and incubated with Lyso-Prime Deep Red working solution (1000X) for 30 min at 37°C. After this, cells were washed again and incubated with pHLys Green working solution (1000X) for an additional 30 min at 37°C. After final washes, a cell growth medium was added with nuclear stain, and the cells were observed under a confocal microscope. The fluorescence intensities of pHLys Green (excitation at 488 nm, emission at 500–600 nm) and LysoPrime Deep Red (excitation at 633 nm, emission at 640–700 nm) were measured on an Andor DragonFly

200 platform using a cf40 Zyla camera attached to a Leica DMi8 confocal microscope using the Fusion software. Images were exported to tiff format and fluorescence intensity ratios calculated using the ROI function in ImageJ.

## Statistical analysis

Data were expressed as means ± SD. N represents independent culture preparations. The statistical significance between the two groups was assessed using the $t$ test. For comparisons involving multiple groups with a single factor, one-way ANOVA was used, followed by Tukey's post hoc test for multiple comparison adjustments. For analyses involving multiple groups with two factors, two-way ANOVA was employed, followed by Tukey's post hoc test for adjustments. A $P$-value of less than 0.05 was considered indicative of statistical significance.

# Data Availability

Datasets reported in this study are not composed of standardized data types. No original code was reported in the study. All data generated or analyzed during this study are included in this published article and its supplementary information files. Any additional information required to reanalyze the data reported in this study is available from the lead contact upon request.

# Supplementary Information

# Acknowledgements

This work was supported by the National Institute of Mental Health (MH119000), National Institute of Mental Health (MH134592), and National Institute on Drug Abuse (DA059280).

## Author Contributions

N Rezagholizadeh: conceptualization, data curation, formal analysis, validation, investigation, methodology, and writing—original draft.
G Datta: formal analysis, investigation, and methodology.
WA Hasler: formal analysis and investigation.
EC Nguon: investigation.
EV Smokey: investigation.
N Khan: formal analysis, investigation, and methodology.
X Chen: conceptualization, data curation, funding acquisition, validation, and writing—review and editing.

## Conflict of Interest Statement

The authors declare that they have no conflict of interest.

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
