## [Reviewer comments · Life Science Alliance]

Life Science Alliance

SLC38A9 is directly involved in Tat-induced endolysosomal dysfunction and senescence in astrocytes

Neda Rezagholizadeh, Gaurav Datta, Wendie Hasler, Erica Nguon, Elise Smokey, Nabab Khan, and Xuesong Chen
DOI: <https://doi.org/10.26508/lsa.202503231>

Corresponding author(s): Xuesong Chen, University of North Dakota

Review Timeline:

Submission Date:	2025-01-21
Editorial Decision:	2025-02-20
Revision Received:	2025-04-02
Editorial Decision:	2025-04-23
Revision Received:	2025-04-25
Accepted:	2025-04-28

Scientific Editor: Tim Fessenden

Transaction Report:

February 20, 2025

Re: Life Science Alliance manuscript #LSA-2025-03231-T

Dr. Xuesong Chen
University of North Dakota
Biomedical Sciences
504 Hamline St
Grand Forks, ND 58203

Dear Dr. Chen,

Thank you for submitting your manuscript entitled "HIV-1 Tat induces cellular senescence via SLC38A9 in human astrocytes" to Life Science Alliance. The manuscript was assessed by expert reviewers, whose comments are appended to this letter. We invite you to submit a revised manuscript addressing the Reviewer comments.

Thank you for this interesting contribution to Life Science Alliance. We are looking forward to receiving your revised manuscript.

Sincerely,

B. MANUSCRIPT ORGANIZATION AND FORMATTING:

Reviewer #1 (Comments to the Authors (Required)):

The manuscript titled "HIV-1 Tat induces cellular senescence via SLC38A9 in human astrocytes" presents compelling evidence that HIV-1 Tat interacts with the endolysosome-resident arginine sensor SLC38A9, leading to endolysosome dysfunction and cellular senescence in astrocytes. The study highlights a novel mechanism by which Tat disrupts astrocytic function and contributes to HIV-associated neurocognitive disorders (HAND). The authors demonstrate that the arginine-rich basic domain of Tat is critical for this interaction, leading to senescence-like changes characterized by increased SA- β -gal activity, elevated p16 and p21 levels, and the release of SASP factors. Moreover, knockdown of SLC38A9 attenuates Tat-induced endolysosome dysfunction, LTR transactivation, and cellular senescence. While the study is well-executed, several key points require clarification and additional experimental validation.

Comments:

1. The study relies on SA- β -gal staining as a primary marker of senescence. Have the authors considered including additional markers such as γ H2AX or telomere-associated foci to further validate cellular senescence?
2. The authors used a 100 nM concentration of Tat for 48 hours. It would be important to understand how this concentration is relevant to clinical settings. Are the concentrations used in the study within the range typically observed in patients with HAND? How do these levels compare to the amount of Tat detected in cerebrospinal fluid (CSF) or brain tissue from affected individuals? Providing references to support the physiological relevance of these Tat concentrations would strengthen the study.
3. The study suggests that silencing SLC38A9 inhibits the release of HIV Tat into the cytosol. What happens to the accumulated Tat protein in the lysosome? Does it get degraded by the lysosomal machinery, or does it remain trapped within the lysosome? Assessing Tat expression in SLC38A9-silenced cells would help clarify this point.
4. The authors discuss the interaction between Tat and SLC38A9 using co-immunoprecipitation and live-cell imaging. However, direct interaction studies such as surface plasmon resonance or isothermal titration calorimetry could provide stronger biophysical evidence.
5. The study claims that endolysosome dysfunction is an early event in Tat-induced senescence, yet a detailed temporal analysis of these changes is missing. Time-course experiments assessing senescence markers alongside endolysosome pH alterations would strengthen this claim.
6. The authors justify a 48-hour time point for the analysis of senescence and a 2-hour time point for mTOR activation. However, how does this timeline relate to Tat's internalization kinetics? Additional clarification on how Tat's intracellular processing aligns with these time points would be beneficial.
7. The functional consequences of Tat-SLC38A9 interaction on astrocyte physiology remain unclear. How does this affect astrocyte-mediated neuronal support, glutamate uptake, or metabolic homeostasis, which are crucial for HAND pathology?
8. The study suggests that SLC38A9 is necessary for Tat-induced senescence. Could SLC38A9-mediated mTORC1 activation lead to SASP induction independent of direct lysosomal damage? How does this relate to existing literature on mTORC1 and senescence?
9. The role of SLC38A9 in Tat-induced senescence and endolysosome dysfunction should be validated in an in vivo model of Tat exposure to confirm the relevance of the findings in a physiological context.
10. The authors provide evidence that SLC38A9 knockdown attenuates Tat-induced cellular senescence. However, have they considered overexpressing SLC38A9 in the absence of Tat to determine whether SLC38A9 itself contributes to senescence? This would help clarify whether the observed effects are solely due to Tat exposure or if SLC38A9 overexpression can independently drive senescence.

11. The study suggests that SLC38A9 knockdown restores normal astrocyte function. To further validate this, have the authors assessed endolysosomal function using additional markers such as LysoTracker, cathepsin B, or LAMP-1? These markers could provide a more comprehensive picture of endolysosome integrity.
12. The manuscript would benefit from additional information about the structure of the HIV Tat protein, particularly its binding motifs and active sites. Specifically, are there other regions beyond the arginine-rich domain that contribute to Tat's toxic effects?
13. More information on the normal physiological function of SLC38A9 would be helpful. Is SLC38A9 exclusively present in lysosomes, or does it have additional cellular functions? Understanding its baseline role in healthy cells would provide crucial context for interpreting its involvement in Tat-induced dysfunction.
14. The link between Tat-induced endolysosome damage and HIV-1 LTR transactivation is intriguing. However, does this also translate into increased viral replication, or is it limited to Tat-mediated LTR activation? Measuring HIV-1 RNA levels would provide further insight into the impact on viral reactivation.
15. The conclusion suggests SLC38A9 as a therapeutic target. Have the authors tested any pharmacological inhibitors or genetic overexpression approaches to further validate this claim?
16. Some references on endolysosomal dysfunction in neurodegeneration and HAND could be expanded to place the findings in a broader context. Additionally, a clearer discussion of whether endolysosomal dysfunction is a cause or consequence of senescence would be beneficial.

Referee Cross-Commenting:

I appreciate the insightful comments from Reviewers #2 and #3. I agree that assessing the long-term effects of Tat-induced senescence and revising the title for clarity would enhance the manuscript. However, I emphasize the need for biophysical validation of the Tat-SLC38A9 interaction experimentally. Additionally, clarifying the fate of Tat in SLC38A9-silenced cells would strengthen mechanistic claims. While lysosomal inhibition studies are valuable, using targeted inhibitors rather than chloroquine may yield more specific insights. Overall, addressing these points would significantly improve the study's impact and mechanistic depth.

Reviewer #2 (Comments to the Authors (Required)):

In the manuscript (LSA-2025-03231-T), Rezagholizadeh and colleagues determined the relationship between HIV-1 Tat-induced endolysosomal damage and senescence in human astrocytes. They first demonstrated that Tat induced senescence in astrocytes in a concentration-dependent manner (0, 1, 10, 100, 200 nM), while the basic domain-deleted Tat mutant did not. They also found that The Tat mutant failed to induce endolysosomal damage. These findings led them to determine the roles of the endolysosome-resident arginine sensor SLC38A9 in the processes. They showed that Tat, but not Tat mutant, formed a complex with SLC38A9 and activated mTOR signaling pathway, and SLC38A9 knockdown abrogated the mTOR signaling activation. They also demonstrated that SLC38A9 knockdown significantly impaired Tat-induced endolysosomal dysfunction, LTR transactivation, and senescence. Thus, they concluded the important roles of SLC38A9 in these processes in human astrocytes.

The study generally consisted of a series of well-designed and thought-out experiments with the outcomes supporting the main conclusion. This is the first report about the roles of SLC38A9 in Tat effects in astrocytes. These findings will add to our understanding of Tat protein in HIV-associated neurocognitive disorders and accelerated aging. The manuscript was well written, and the data were clearly presented.

Just one suggestion for the authors to consider: To precisely convey the findings presented in the manuscript, I suggest changing the manuscript title to "SLC38A9 is directly involved in Tat-induced endolysosomal dysfunction and senescence in human astrocytes."

Reviewer #3 (Comments to the Authors (Required)):

In this manuscript the author's address a critically important question relating to senescence and hence aging of the brain in the context of HIV infection. Through a series of very well designed experiments and they demonstrate that the tat protein of HIV is capable of inducing senescence in astrocytes. This is a highly impactful and significant observation since it has been demonstrated that HIV infected individuals even on and retroviral therapy lack transcriptional control and release date into the extracellular space from infected cells that has the opportunity to interact with astrocytes in the brain. In this manuscript they map out the pathway as to how that is internalized how it interacts with the lysosomes and eventually leads to senescence. The experiments are well designed with proper controls and the pathway has been discerned in a step wise approach. I have a few suggestions for improving the work.

1. Could consider using chloroquine to inhibit the lysosomes and see how that may affect the senescence mediated by tat.
2. The effect of that in most experiments is assessed at a maximum of 48 hours 1 wonders if this is a long lasting effect or could this all be transient further along term experiments may be helpful in this regard.
3. Endolysosome is misspelled in a couple of places.

Response to reviewers

Reviewer #1:

The manuscript titled "HIV-1 Tat induces cellular senescence via SLC38A9 in human astrocytes" presents compelling evidence that HIV-1 Tat interacts with the endolysosome-resident arginine sensor SLC38A9, leading to endolysosome dysfunction and cellular senescence in astrocytes. The study highlights a novel mechanism by which Tat disrupts astrocytic function and contributes to HIV-associated neurocognitive disorders (HAND). The authors demonstrate that the arginine-rich basic domain of Tat is critical for this interaction, leading to senescence-like changes characterized by increased SA- β -gal activity, elevated p16 and p21 levels, and the release of SASP factors. Moreover, knockdown of SLC38A9 attenuates Tat-induced endolysosome dysfunction, LTR transactivation, and cellular senescence. While the study is well-executed, several key points require clarification and additional experimental validation.

Response: We thank the reviewer for the positive comments

Comments:

1. The study relies on SA- β -gal staining as a primary marker of senescence. Have the authors considered including additional markers such as γ H2AX or telomere-associated foci to further validate cellular senescence?

Response: In the present study, we have used three sets of markers to demonstrate Tat-induced cellular senescence, including increased SA- β -gal activity, elevated p16 and p21 levels, and the release of SASP factors (IL-6, IL-8, and CCL2).

We also assessed the extent to which Tat affects cellular proliferation using a BrdU incorporation assay; BrdU will be incorporated into newly synthesized DNA of actively proliferating cells. We demonstrated that Tat, but not mutant Tat lacking the arginine-rich domain, significantly reduced BrdU incorporation, indicating that Tat induces the inhibition of DNA replication and/or cellular proliferation in astrocytes. However, such data was used in another independent study.

Nonetheless, we did not determine every marker of cellular senescence; However, we would like to assess other markers of cellular senescence, such as γ H2AX and telomere-associated foci, to investigate whether and how Tat-induced endolysosome dysfunction could lead to DNA damage in the nucleus in the future.

2. The authors used a 100 nM concentration of Tat for 48 hours. It would be important to understand how this concentration is relevant to clinical settings. Are the concentrations

used in the study within the range typically observed in patients with HAND? How do these levels compare to the amount of Tat detected in cerebrospinal fluid (CSF) or brain tissue from affected individuals? Providing references to support the physiological relevance of these Tat concentrations would strengthen the study.

Response: Although Tat concentrations in brain parenchyma are unknown, nanomolar concentrations of Tat has been detected in CSF of HIV infected individuals on ART drugs ^{1,2}, thus local concentrations of Tat in brain parenchyma could be quite high. In the present study, we did observe that Tat at lower concentration (10 nM for 48 hr) significantly increased the percentage of SA- β -gal positive cells. Such physiological relevant evidence is now included in the manuscript.

The reason why we chose to use higher concentration of Tat for most of the experiments in the study is because Tat at the concentration of 100 nM induces robust effects on endolysosome dysfunction and cellular senescence. With such robust effects, we can confidently assess whether silencing SLC38A9 could attenuate Tat-induced endolysosome dysfunction and cellular senescence.

3. The study suggests that silencing SLC38A9 inhibits the release of HIV Tat into the cytosol. What happens to the accumulated Tat protein in the lysosome? Does it get degraded by the lysosomal machinery, or does it remain trapped within the lysosome? Assessing Tat expression in SLC38A9-silenced cells would help clarify this point.

Response: In a published study conducted in astrocytes ³, we have demonstrated internalized Tat remains trapped in endolysosomes for extended periods of time (up to 96 hours) without being fully degraded. This is, in part, because Tat induces endolysosome de-acidification and dysfunction, which impairs the degradation capability of endolysosomes.

Since silencing SLC38A9 attenuates Tat-induced endolysosome dysfunction, we have assessed Tat expression in SLC38A9-silenced cells (as suggested). We found that silencing SLC38A9 decreased cellular Tat levels, which indicates that silencing SLC38A9 enhances Tat degradation. Such data is now included as Figure 4D.

4. The authors discuss the interaction between Tat and SLC38A9 using co-immunoprecipitation and live-cell imaging. However, direct interaction studies such as surface plasmon resonance or isothermal titration calorimetry could provide stronger biophysical evidence.

Response: In the present study, we have the following evidence supporting the interaction between Tat and SLC38A9. (1) Findings from co-immunoprecipitation studies from both directions demonstrate the interaction between Tat and SLC38A9. (2) In a co-immunoprecipitation study using mutant Tat lacking the arginine-rich domain, we demonstrated that arginine-rich domain of Tat could

interact with SLC38A9. (3) Co-localization study conducted in live cells indicate that Tat could interact with SLC38A9 at the site of endolysosome. (4) Functional study showing that Tat activates mTORC1, which is blocked by SLC38A knockdown, further supports the interaction between Tat and SLC38A9, because SLC38A9 is as a known activator of mTORC1⁴⁻⁸.

We agree with the reviewer that direct interaction studies such as surface plasmon resonance or isothermal titration calorimetry could provide biophysical evidence to further support the interaction between Tat and SLC38A9. However, we feel that findings from such studies may not provide conclusive evidence showing that Tat interacts with SLC38A9 (an integral membrane protein) in the lumen of endolysosome with the presence of intact membrane. Furthermore, findings from such studies may not provide substantial insights into interaction between Tat and SLC38A9 at molecular level. We feel that the interaction between Tat and SLC38A9 at biophysical and molecular level warrants further investigation in an independent paper.

5. The study claims that endolysosome dysfunction is an early event in Tat-induced senescence, yet a detailed temporal analysis of these changes is missing. Time-course experiments assessing senescence markers alongside endolysosome pH alterations would strengthen this claim.

Response: In the present study, we have shown Tat enters endolysosomes 1 hr following the treatment of exogenous Tat. We have further shown that Tat (100 nM, 2 hr and 24 hr) induced endolysosome damage as indicated by the formation of GFP-galetin-3 puncta. In a published study³, we also have shown that Tat (100 nM for 18 hr) increased endolysosome pH, enlarged endolysosomes, and decreased endolysosome enzyme activities in human primary astrocytes.

In a time course study, we have demonstrated that Tat (100nM) significantly increased release of SASP factor (IL-6) into the extracellular media of primary human astrocytes at 48 hours post-treatment, but not at earlier time points, and such findings are now included as Figure 1C. Such findings support our claim that endolysosome dysfunction is an early event in Tat-induced senescence, and we have made this point clear in the revised manuscript.

6. The authors justify a 48-hour time point for the analysis of senescence and a 2-hour time point for mTOR activation. However, how does this timeline relate to Tat's internalization kinetics? Additional clarification on how Tat's intracellular processing aligns with these time points would be beneficial.

Response: In the present study, we have shown that Tat enters endolysosomes 1 hr following the treatment of exogenous Tat, and that Tat (100 nM) induced endolysosome damage as early as 2 hr post-treatment. Thus, 2-hour time point for mTOR activation aligns with Tat's internalization kinetics, and we have made this point clear in the revised manuscript.

7. The functional consequences of Tat-SLC38A9 interaction on astrocyte physiology remain unclear. How does this affect astrocyte-mediated neuronal support, glutamate uptake, or metabolic homeostasis, which are crucial for HAND pathology?

Response: As the reviewer mentioned, astrocytes are critical for CNS physiology by providing neuronal support, facilitating synaptic signaling, and maintaining metabolic homeostasis. As such disrupting astrocyte functions could lead to neurodegeneration⁹⁻¹³, which contributes to HAND pathology.

In the present study, we only focused on how Tat-SLC38A9 interaction affects astrocyte function (endolysosome function and cellular senescence), and we have not yet explored the consequence of how such astrocyte dysfunction may affect other CNS cells. Because, endolysosomes in astrocytes play a critical role in maintaining a healthy nervous system¹⁴, and endolysosome dysfunction in astrocytes alone leads to neurodegeneration¹⁵, Tat-induced endolysosome dysfunction in astrocytes could lead to neurodegeneration. Thus, we do plan to address how such Tat-induced astrocyte dysfunction may affect other CNS cells in future studies. We have made this point clear in the revised manuscript.

8. The study suggests that SLC38A9 is necessary for Tat-induced senescence. Could SLC38A9-mediated mTORC1 activation lead to SASP induction independent of direct lysosomal damage? How does this relate to existing literature on mTORC1 and senescence?

Response: mTORC1 activation has been shown to drive many senescence-like phenotypes^{16,17}. Thus, it is possible that SLC38A9-mediated mTORC1 activation could lead to SASP induction independent of direct lysosomal damage.

However, a recent study has shown that mTORC1 activation induces disassembly of v-ATPase on endolysosomes, thus impairing the degradation capability of endolysosomes. Could mTORC1 activation also lead to endolysosome damage? Although this is an open question, the proteolipid ring (c-ring) of V₀ sector of v-ATPase can form a protein core with a diameter of 3.5 nm¹⁸, which opens in the presence of calcium¹⁹. Thus, it is possible that mTORC1 activation could lead to endolysosome membrane leakage, which could then contribute to cellular senescence²⁰. On the other hand, endolysosome membrane leakage could lead to inhibition of mTORC1²¹ likely as a protective mechanism. Thus, the question of whether SLC38A9-mediated mTORC1 activation leads to SASP induction independent of direct lysosomal damage warrants further investigation. We have added these statements in the revised manuscript

9. The role of SLC38A9 in Tat-induced senescence and endolysosome dysfunction should be validated in an in vivo model of Tat exposure to confirm the relevance of the

findings in a physiological context.

Response: We agree that role of SLC38A9 in Tat-induced senescence and endolysosome dysfunction should be validated in an in vivo model of Tat exposure. However, SLC38A9 knockout animals is not currently available, which limits our progression on in vivo studies. But we do plan to develop SLC38A9 knockout or conditioned SLC38A9 knockout mice and conduct such in vivo study in the future.

10. The authors provide evidence that SLC38A9 knockdown attenuates Tat-induced cellular senescence. However, have they considered overexpressing SLC38A9 in the absence of Tat to determine whether SLC38A9 itself contributes to senescence? This would help clarify whether the observed effects are solely due to Tat exposure or if SLC38A9 overexpression can independently drive senescence.

Response: We have assessed the extent to which Tat treatment affects the expression of SLC38A9 using immunoblotting methods, and we found that Tat did not induce significant changes in the expression of SLC38A9. Thus, we were not compelled to test if SLC38A9 overexpression can drive senescence. Along the line of the reviewer's comment, we would like to assess whether activating SLC38A9 without Tat could drive senescence; However, no such pharmacological activators are available.

11. The study suggests that SLC38A9 knockdown restores normal astrocyte function. To further validate this, have the authors assessed endolysosomal function using additional markers such as LysoTracker, cathepsin B, or LAMP-1? These markers could provide a more comprehensive picture of endolysosome integrity.

Response: We have evidence that SLC38A9 knockdown attenuates Tat-induced release of cathepsin B and galectin 3, indicating that SLC38A9 knockdown attenuates Tat-induced endolysosome dysfunction.

As suggested, we have conduct additional studies to assess whether SLC38A9 knockdown restores endolysosome function. We demonstrated that SLC38A9 knockdown attenuates Tat-induced endolysosome membrane leakage, as indicated by galectin-3 puncta formation in LAMP-1 positive vesicles. Such data is now included as Figure 4C.

12. The manuscript would benefit from additional information about the structure of the HIV Tat protein, particularly its binding motifs and active sites. Specifically, are there other regions beyond the arginine-rich domain that contribute to Tat's toxic effects?

Response: Beside the neurotoxic basic arginine-rich domain (49-57)²²⁻²⁶, Tat is composed of several other domains, with distinct functions crucial for viral replication and pathogenesis. Proline-rich domain (1-20) is important for stabilizing Tat's binding to the inner leaf of the cell membrane²⁷; Cysteine-rich

domain (21-37) is important for dimerization and metal binding, playing a critical role in the activation of HIV genomic DNA transcription²⁸, and it is also involved in the binding of Tat to TLR4²⁹ and the CXCR4³⁰⁻³²; Core domain (38-48) is important for binding to the CDK9-associated C-type cyclin, which is crucial for Tat's transactivation activity²⁸, and it is also involved in the interaction of Tat with LRP1³³⁻³⁵. Glutamine-Rich Domain (58-71) plays a role in interacting with the TAR region of HIV RNA, and additionally it is implicated in Tat-mediated apoptosis³⁶; RGD Domain (72-85) is crucial for its interaction with integrins, which is important for Tat's involvement of cellular adhesion and signaling processes³⁷⁻⁴²; C-terminal domain (86-101) is important for NF-κB activity⁴³, a crucial factor in regulating immune responses and inflammation.

Although these regions of Tat could induce inflammatory responses and other toxic effects, our findings suggest that the arginine-rich domain is critical for Tat-induced senescence-like phenotype, since mutant Tat lacking the arginine-rich basic domain failed to induce senescence-like phenotype. We have included this information in the revised version.

13. More information on the normal physiological function of SLC38A9 would be helpful. Is SLC38A9 exclusively present in lysosomes, or does it have additional cellular functions? Understanding its baseline role in healthy cells would provide crucial context for interpreting its involvement in Tat-induced dysfunction.

Response: Belongs to solute carrier (SLC) family, SLC38A9 is an amino acid transporter^{7,44}, competent for transporting glutamine, leucine, and arginine. SLC38A9 is an endolysosomal membrane proteins, with eleven transmembrane helices⁴⁵. SLC38A9 interacts with the Rag-Regulator complex to activate mTORC1⁴⁴, and it has been shown that SLC38A9 signals arginine sufficiency in the lumen of endolysosomes, which is critical for regulating the activity of mTORC1⁴⁻⁸. Independent of arginine sensing, lysosomal cholesterol has been shown to drive mTORC1 activation through the SLC38A9-NPC1 complex⁴⁶. Thus, SLC38A9 encompasses the functions of both a transporter and a receptor, but the signaling may not involve amino acid transport⁴⁷. We have included this information in the revised version.

14. The link between Tat-induced endolysosome damage and HIV-1 LTR transactivation is intriguing. However, does this also translate into increased viral replication, or is it limited to Tat-mediated LTR activation? Measuring HIV-1 RNA levels would provide further insight into the impact on viral reactivation.

Response: In a published study³, we have demonstrated that extracellular Tat enters astrocytes via endocytosis, that Tat accumulated in endolysosomes is functionally intact, and that upon release from endolysosomes, Tat induces HIV-1 LTR transactivation. Thus, internalized extracellular Tat could play an important role in latent infection of HIV-1, and clearing endolysosome Tat represents an additional target of therapeutic strategies. Consistent with this notion, it has been

shown that autophagy restricts HIV-1 infection by selectively degrading Tat in CD4+ T lymphocytes⁴⁸.

We agree with the reviewer that measuring HIV-1 RNA levels would provide further insight into the impact on viral reactivation; However, in our system, cells were stably express HIV-1 LTR with a luciferase reporter gene. In the future, we would like to investigate further whether and how internalized Tat could affect HIV-1 RNA levels and HIV-replication using HIV-1 latency models.

15. The conclusion suggests SLC38A9 as a therapeutic target. Have the authors tested any pharmacological inhibitors or genetic overexpression approaches to further validate this claim?

Response: SLC38A9 is newly discovered endolysosomal amino acid transporter, and currently no pharmacological activator or inhibitor is available. SLC38A9 knockout animals is also not currently available. As addressed above, we do plan to develop SLC38A9 knockout or conditioned SLC38A9 knockout mice to further validate the claim that SLC38A9 is a therapeutic target in the future.

16. Some references on endolysosomal dysfunction in neurodegeneration and HAND could be expanded to place the findings in a broader context. Additionally, a clearer discussion of whether endolysosomal dysfunction is a cause or consequence of senescence would be beneficial.

Response: As suggested, we have expanded the references on endolysosomal dysfunction in neurodegeneration and HAND.

Endolysosome dysfunction could lead to abnormal accumulation of undegraded materials (macromolecules and mitochondria) in endolysosomes and endolysosome enlargement^{3,49}, mitochondrial dysfunction⁵⁰⁻⁵², impaired clearance of viral factors^{3,53}, augmented release of their luminal contents via exocytosis^{54,55} that contribute to inflammation⁵⁶⁻⁶⁰, and synaptodendritic impairment⁶¹. As such, dysfunction of endolysosome contributes to the development of neurodegeneration disorders including AD^{62,63}, PD⁶⁴, ALS⁶⁵, and HAND⁶⁶.

We also stated clearly that endolysosome dysfunction represents as a cause of cellular senescence.

Referee Cross-Commenting:

I appreciate the insightful comments from Reviewers #2 and #3. I agree that assessing the long-term effects of Tat-induced senescence and revising the title for clarity would enhance the manuscript. However, I emphasize the need for biophysical validation of the Tat-SLC38A9 interaction experimentally. Additionally, clarifying the fate of Tat in SLC38A9-silenced cells would strengthen mechanistic claims. While lysosomal

inhibition studies are valuable, using targeted inhibitors rather than chloroquine may yield more specific insights. Overall, addressing these points would significantly improve the study's impact and mechanistic depth.

Response: Longer-term effects of Tat-induced senescence are now included, and we have revised the title as suggested.

With regards to Tat-SLC38A9 interaction, we have the following evidence supporting the interaction between Tat and SLC38A9 in the present study. (1) Findings from co-immunoprecipitation studies from both directions demonstrate the interaction between Tat and SLC38A9. (2) In a co-immunoprecipitation study using mutant Tat lacking the arginine-rich domain, we demonstrated that arginine-rich domain of Tat could interact with SLC38A9. (3) Co-localization study conducted in live cells indicate that Tat could interact with SLC38A9 at the site of endolysosome. (4) Functional study showing that Tat activates mTORC1, which is blocked by SLC38A knockdown, further supports the interaction between Tat and SLC38A9, because SLC38A9 is as a known activator of mTORC1 ⁴⁻⁸.

We agree with the reviewer #1 that direct interaction studies such as surface plasmon resonance or isothermal titration calorimetry could provide biophysical evidence to further support the interaction between Tat and SLC38A9. However, we feel that findings from such studies may not provide conclusive evidence showing that Tat interacts with SLC38A9 (an integral membrane protein) in the lumen of endolysosome with the presence of intact membrane. Furthermore, findings from such studies may not provide substantial insights into interaction between Tat and SLC38A9 at molecular level. We feel that the interaction between Tat and SLC38A9 at biophysical and molecular level warrants further investigation in an independent paper.

As suggested by reviewer #1, we have assessed the function of endolysosomes and the fate of Tat in SLC38A9-silenced cells, and we demonstrated that SLC38A9 knockdown attenuates Tat-induced endolysosome membrane leakage, as indicated by galectin-3 puncta formation. Furthermore, we found that silencing SLC38A9 decreased cellular Tat levels, which indicates that silencing SLC38A9 enhance Tat degradation.

Regarding further studies using specific lysosomal inhibitors as mentioned by reviewer #3, it has been shown that inhibiting endolysosome degradation exacerbates the phenotypes of senescence ⁶⁷. Silencing transcription factor EB (TFEB), which functions as a master regulator of the autophagy-lysosome pathway, also exacerbates senescence ²⁰. And we have added these references to the revised manuscript. In the field of NeuroHIV, we have shown that many other HIV-factors de-acidify endolysosome and inhibit endolysosomes degradation including gp120 ^{55,68,69}, weak basic antiretroviral drugs ⁷⁰, and drug of abuse (methamphetamine, preliminary data), we would like that conduct possible additive or synergistic effects of these HIV-factors on Tat-induced

cellular senescence in future studies using specific inhibitor of endolysosomes as positive controls such as v-ATPase inhibitor bafilomycin, and LLOMe that enters endolysosome lumen and specifically induces lysosomal membrane permeabilization.

Reviewer #2

In the manuscript (LSA-2025-03231-T), Rezagholizadeh and colleagues determined the relationship between HIV-1 Tat-induced endolysosomal damage and senescence in human astrocytes. They first demonstrated that Tat induced senescence in astrocytes in a concentration-dependent manner (0, 1, 10, 100, 200 nM), while the basic domain-deleted Tat mutant did not. They also found that The Tat mutant failed to induce endolysosomal damage. These findings led them to determine the roles of the endolysosome-resident arginine sensor SLC38A9 in the processes. They showed that Tat, but not Tat mutant, formed a complex with SLC38A9 and activated mTOR signaling pathway, and SLC38A9 knockdown abrogated the mTOR signaling activation. They also demonstrated that SLC38A9 knockdown significantly impaired Tat-induced endolysosomal dysfunction, LTR transactivation, and senescence. Thus, they concluded the important roles of SLC38A9 in these processes in human astrocytes.

The study generally consisted of a series of well-designed and thought-out experiments with the outcomes supporting the main conclusion. This is the first report about the roles of SLC38A9 in Tat effects in astrocytes. These findings will add to our understanding of Tat protein in HIV-associated neurocognitive disorders and accelerated aging. The manuscript was well written, and the data were clearly presented.

Just one suggestion for the authors to consider: To precisely convey the findings presented in the manuscript, I suggest changing the manuscript title to "SLC38A9 is directly involved in Tat-induced endolysosomal dysfunction and senescence in human astrocytes."

Response: We thank the reviewer for the positive comments. As suggested, we have now changed the title to "SLC38A9 is directly involved in Tat-induced endolysosomal dysfunction and senescence in astrocytes".

Reviewer #3:

In this manuscript the author's address a critically important question relating to senescence and hence aging of the brain in the context of HIV infection. Through a series of very well-designed experiments and they demonstrate that the tat protein of HIV is capable of inducing senescence in astrocytes. This is a highly impactful and significant observation since it has been demonstrated that HIV infected individuals even on and retroviral therapy lack transcriptional control and release date into the

extracellular space from infected cells that has the opportunity to interact with astrocytes in the brain. In this manuscript they map out the pathway as to how that is internalized how it interacts with the lysosomes and eventually leads to senescence. The experiments are well designed with proper controls and the pathway has been discerned in a step wise approach. I have a few suggestions for improving the work.

Response: We thank the reviewer for the positive comments.

1. Could consider using chloroquine to inhibit the lysosomes and see how that may affect the senescence mediated by tat.

Response: We thank the review for the suggestion. It has been shown that inhibiting endolysosome degradation exacerbates the phenotypes of senescence⁶⁷. Silencing transcription factor EB (TFEB), which functions as a master regulator of the autophagy–lysosome pathway, also exacerbates senescence²⁰. This evidence indicates that inhibiting endolysosome function could drive the development of senescence. We have added these references to the revised manuscript.

In the field of NeuroHIV, we have shown that many other HIV-factors de-acidify endolysosomes and induces endolysosomes dysfunction, including gp120, weak basic antiretroviral drugs, and drug of abuse (methamphetamine), we would like that conduct possible additive or synergistic interactions of these HIV-factors on Tat-induced cellular senescence in future studies using specific inhibitor of endolysosomes as positive controls, such as v-ATPase inhibitor bafilomycin, and LLOMe that enters endolysosome lumen and specifically induces lysosomal membrane permeabilization.

2. The effect of that in most experiments is assessed at a maximum of 48 hours 1 wonders if this is a long-lasting effect or could this all be transient further along term experiments may be helpful in this regard.

Response: We did observe that Tat treatment for 72 hr also induced the development of senescence-like phenotype, which is now included as Figure 1D in the revised manuscript.

3. Endolysosome is misspelled in a couple of places.

Response: We have corrected those misspelled words.

Reference

- 1 Johnson, T. P. *et al.* Induction of IL-17 and nonclassical T-cell activation by HIV-Tat protein. *Proc Natl Acad Sci U S A* **110**, 13588-13593, doi:1308673110 [pii] 10.1073/pnas.1308673110 (2013).
- 2 Henderson, L. J. *et al.* Presence of Tat and transactivation response element in spinal fluid despite antiretroviral therapy. *Aids* **33**, S145-S157 (2019).
- 3 Khan, N. *et al.* HIV-1 Tat endocytosis and retention in endolysosomes affects HIV-1 Tat-induced LTR transactivation in astrocytes. *FASEB J* **36**, e22184, doi:10.1096/fj.202101722R (2022).
- 4 Wang, S. *et al.* Lysosomal amino acid transporter SLC38A9 signals arginine sufficiency to mTORC1. *Science* **347**, 188-194 (2015).
- 5 Wang, S. *et al.* The amino acid transporter SLC38A9 is a key component of a lysosomal membrane complex that signals arginine sufficiency to mTORC1. *Science (New York, NY)* **347**, 188 (2015).
- 6 Jung, J., Genau, H. M. & Behrends, C. Amino acid-dependent mTORC1 regulation by the lysosomal membrane protein SLC38A9. *Molecular and cellular biology* (2015).
- 7 Rebsamen, M. *et al.* SLC38A9 is a component of the lysosomal amino acid sensing machinery that controls mTORC1. *Nature* **519**, 477-481, doi:10.1038/nature14107 (2015).
- 8 Lei, H.-T., Mu, X., Hattne, J. & Gonen, T. A conformational change in the N terminus of SLC38A9 signals mTORC1 activation. *Structure* **29**, 426-432. e428 (2021).
- 9 Dickens, A. M. *et al.* Chronic low-level expression of HIV-1 Tat promotes a neurodegenerative phenotype with aging. *Sci Rep* **7**, 7748, doi:10.1038/s41598-017-07570-5 (2017).
- 10 Zhao, X. *et al.* Long-term HIV-1 Tat Expression in the Brain Led to Neurobehavioral, Pathological, and Epigenetic Changes Reminiscent of Accelerated Aging. *Aging Dis* **11**, 93-107, doi:10.14336/AD.2019.0323 (2020).
- 11 Cole, J. H. *et al.* Increased brain-predicted aging in treated HIV disease. *Neurology* **88**, 1349-1357, doi:10.1212/WNL.0000000000003790 (2017).
- 12 Mackiewicz, M. M., Overk, C., Achim, C. L. & Masliah, E. Pathogenesis of age-related HIV neurodegeneration. *J Neurovirol* **25**, 622-633, doi:10.1007/s13365-019-00728-z (2019).
- 13 Zhao, X., Zhang, F., Kandel, S. R., Brau, F. & He, J. J. HIV Tat and cocaine interactively alter genome-wide DNA methylation and gene expression and exacerbate learning and memory impairments. *Cell Rep* **39**, 110765, doi:10.1016/j.celrep.2022.110765 (2022).
- 14 Kreher, C., Favret, J., Maulik, M. & Shin, D. Lysosomal Functions in Glia Associated with Neurodegeneration. *Biomolecules* **11**, doi:10.3390/biom11030400 (2021).
- 15 Di Malta, C., Fryer, J. D., Settembre, C. & Ballabio, A. Astrocyte dysfunction triggers neurodegeneration in a lysosomal storage disorder. *Proc Natl Acad Sci U S A* **109**, E2334-2342, doi:10.1073/pnas.1209577109 (2012).

- 16 Herranz, N. *et al.* mTOR regulates MAPKAPK2 translation to control the senescence-associated secretory phenotype. *Nat Cell Biol* **17**, 1205-1217, doi:10.1038/ncb3225 (2015).
- 17 Laberge, R. M. *et al.* MTOR regulates the pro-tumorigenic senescence-associated secretory phenotype by promoting IL1A translation. *Nat Cell Biol* **17**, 1049-1061, doi:10.1038/ncb3195 (2015).
- 18 Couoh-Cardel, S., Hsueh, Y. C., Wilkens, S. & Movileanu, L. Yeast V-ATPase Proteolipid Ring Acts as a Large-conductance Transmembrane Protein Pore. *Sci Rep* **6**, 24774, doi:10.1038/srep24774 (2016).
- 19 Peters, C. *et al.* Trans-complex formation by proteolipid channels in the terminal phase of membrane fusion. *Nature* **409**, 581-588, doi:10.1038/35054500 (2001).
- 20 Suzuki, Y. *et al.* Premature senescence is regulated by crosstalk among TFEB, the autophagy lysosomal pathway and ROS derived from damaged mitochondria in NaAsO(2)-exposed auditory cells. *Cell Death Discov* **10**, 382, doi:10.1038/s41420-024-02139-4 (2024).
- 21 Jia, J. *et al.* Galectins Control mTOR in Response to Endomembrane Damage. *Mol Cell* **70**, 120-135 e128, doi:10.1016/j.molcel.2018.03.009 (2018).
- 22 Tyagi, M., Rusnati, M., Presta, M. & Giacca, M. Internalization of HIV-1 tat requires cell surface heparan sulfate proteoglycans. *Journal of Biological Chemistry* **276**, 3254-3261 (2001).
- 23 Ruiz, A. P. *et al.* A naturally occurring polymorphism in the HIV-1 tat basic domain inhibits uptake by bystander cells and leads to reduced neuroinflammation. *Scientific reports* **9**, 3308 (2019).
- 24 Hui, L., Chen, X., Haughey, N. J. & Geiger, J. D. Role of endolysosomes in HIV-1 Tat-induced neurotoxicity. *ASN neuro* **4**, AN20120017 (2012).
- 25 Sabatier, J. *et al.* Evidence for neurotoxic activity of tat from human immunodeficiency virus type 1. *Journal of virology* **65**, 961-967 (1991).
- 26 Buscemi, L., Ramonet, D. & Geiger, J. D. Human immunodeficiency virus type-1 protein Tat induces tumor necrosis factor- α -mediated neurotoxicity. *Neurobiology of disease* **26**, 661-670 (2007).
- 27 Rayne, F. *et al.* Phosphatidylinositol-(4, 5)-bisphosphate enables efficient secretion of HIV-1 Tat by infected T-cells. *The EMBO journal* **29**, 1348-1362 (2010).
- 28 Wei, P., Garber, M. E., Fang, S.-M., Fischer, W. H. & Jones, K. A. A novel CDK9-associated C-type cyclin interacts directly with HIV-1 Tat and mediates its high-affinity, loop-specific binding to TAR RNA. *Cell* **92**, 451-462 (1998).
- 29 Ben Haij, N. *et al.* HIV-1 Tat protein induces production of proinflammatory cytokines by human dendritic cells and monocytes/macrophages through engagement of TLR4-MD2-CD14 complex and activation of NF- κ B pathway. *PLoS one* **10**, e0129425 (2015).
- 30 Ghezzi, S. *et al.* Inhibition of CXCR4-dependent HIV-1 infection by extracellular HIV-1 Tat. *Biochemical and biophysical research communications* **270**, 992-996 (2000).

- 31 Xiao, H. *et al.* Selective CXCR4 antagonism by Tat: implications for in vivo expansion of coreceptor use by HIV-1. *Proceedings of the National Academy of Sciences* **97**, 11466-11471 (2000).
- 32 Secchiero, P., Zella, D., Capitani, S., Gallo, R. C. & Zauli, G. Extracellular HIV-1 tat protein up-regulates the expression of surface CXC-chemokine receptor 4 in resting CD4+ T cells. *The Journal of Immunology* **162**, 2427-2431 (1999).
- 33 Cafaro, A. *et al.* Role of HIV-1 Tat Protein Interactions with Host Receptors in HIV Infection and Pathogenesis. *International Journal of Molecular Sciences* **25**, 1704 (2024).
- 34 Chen, Y. *et al.* HIV-1 tat regulates occludin and A β transfer receptor expression in brain endothelial cells via Rho/ROCK Signaling pathway. *Oxidative Medicine and Cellular Longevity* **2016**, 4196572 (2016).
- 35 Liu, Y. *et al.* Uptake of HIV-1 tat protein mediated by low-density lipoprotein receptor-related protein disrupts the neuronal metabolic balance of the receptor ligands. *Nature medicine* **6**, 1380-1387 (2000).
- 36 King, J., Eugenin, E., Buckner, C. & Berman, J. HIV tat and neurotoxicity. *Microbes and infection* **8**, 1347-1357 (2006).
- 37 Cafaro, A. *et al.* HIV-1 tat protein enters dysfunctional endothelial cells via integrins and renders them permissive to virus replication. *International journal of molecular sciences* **22**, 317 (2020).
- 38 Monini, P. *et al.* HIV-1 tat promotes integrin-mediated HIV transmission to dendritic cells by binding Env spikes and competes neutralization by anti-HIV antibodies. *PLoS One* **7**, e48781 (2012).
- 39 Urbinati, C. *et al.* Integrin $\alpha v \beta 3$ as a target for blocking HIV-1 Tat-induced endothelial cell activation in vitro and angiogenesis in vivo. *Arteriosclerosis, thrombosis, and vascular biology* **25**, 2315-2320 (2005).
- 40 Urbinati, C. *et al.* $\alpha v \beta 3$ -integrin-dependent activation of focal adhesion kinase mediates NF- κB activation and motogenic activity by HIV-1 Tat in endothelial cells. *Journal of cell science* **118**, 3949-3958 (2005).
- 41 Brake, D. A., Debouck, C. & Biesecker, G. Identification of an Arg-Gly-Asp (RGD) cell adhesion site in human immunodeficiency virus type 1 transactivation protein, tat. *The Journal of cell biology* **111**, 1275-1281 (1990).
- 42 Barillari, G., Gendelman, R., Gallo, R. C. & Ensoli, B. The Tat protein of human immunodeficiency virus type 1, a growth factor for AIDS Kaposi sarcoma and cytokine-activated vascular cells, induces adhesion of the same cell types by using integrin receptors recognizing the RGD amino acid sequence. *Proceedings of the National Academy of Sciences* **90**, 7941-7945 (1993).
- 43 Li, Y., Liu, X., Fujinaga, K., Gross, J. D. & Frankel, A. D. Enhanced NF-kappaB activation via HIV-1 Tat-TRAF6 cross-talk. *Sci Adv* **10**, eadi4162, doi:10.1126/sciadv.adi4162 (2024).
- 44 Schioth, H. B., Roshanbin, S., Hagglund, M. G. & Fredriksson, R. Evolutionary origin of amino acid transporter families SLC32, SLC36 and SLC38 and physiological, pathological and therapeutic aspects. *Mol Aspects Med* **34**, 571-585, doi:10.1016/j.mam.2012.07.012 (2013).

- 45 Lei, H. T., Ma, J., Sanchez Martinez, S. & Gonen, T. Crystal structure of arginine-bound lysosomal transporter SLC38A9 in the cytosol-open state. *Nat Struct Mol Biol* **25**, 522-527, doi:10.1038/s41594-018-0072-2 (2018).
- 46 Castellano, B. M. *et al.* Lysosomal cholesterol activates mTORC1 via an SLC38A9-Niemann-Pick C1 signaling complex. *Science* **355**, 1306-1311, doi:10.1126/science.aag1417 (2017).
- 47 Lei, H. T., Mu, X., Hattne, J. & Gonen, T. A conformational change in the N terminus of SLC38A9 signals mTORC1 activation. *Structure* **29**, 426-432 e428, doi:10.1016/j.str.2020.11.014 (2021).
- 48 Sagnier, S. *et al.* Autophagy restricts HIV-1 infection by selectively degrading Tat in CD4+ T lymphocytes. *J Virol* **89**, 615-625, doi:10.1128/JVI.02174-14 (2015).
- 49 Datta, G. *et al.* SARS-CoV-2 S1 Protein Induces Endolysosome Dysfunction and Neuritic Dystrophy. *Front Cell Neurosci* **15**, 777738, doi:10.3389/fncel.2021.777738 (2021).
- 50 Stepien, K. M. *et al.* Mechanisms of Mitochondrial Dysfunction in Lysosomal Storage Disorders: A Review. *J Clin Med* **9**, doi:10.3390/jcm9082596 (2020).
- 51 Tintos-Hernandez, J. A., Santana, A., Keller, K. N. & Ortiz-Gonzalez, X. R. Lysosomal dysfunction impairs mitochondrial quality control and is associated with neurodegeneration in TBCK encephaloneuronopathy. *Brain Commun* **3**, fcab215, doi:10.1093/braincomms/fcab215 (2021).
- 52 Deus, C. M., Yambire, K. F., Oliveira, P. J. & Raimundo, N. Mitochondria-Lysosome Crosstalk: From Physiology to Neurodegeneration. *Trends Mol Med* **26**, 71-88, doi:10.1016/j.molmed.2019.10.009 (2020).
- 53 Li, S. *et al.* Ambient atmospheric PM worsens mouse lung injury induced by influenza A virus through lysosomal dysfunction. *Respir Res* **24**, 306, doi:10.1186/s12931-023-02618-9 (2023).
- 54 Kim, Y. H. *et al.* Secretory autophagy machinery and vesicular trafficking are involved in HMGB1 secretion. *Autophagy* **17**, 2345-2362, doi:10.1080/15548627.2020.1826690 (2021).
- 55 Datta, G., Miller, N. M., Afghah, Z., Geiger, J. D. & Chen, X. HIV-1 gp120 Promotes Lysosomal Exocytosis in Human Schwann Cells. *Front Cell Neurosci* **13**, 329, doi:10.3389/fncel.2019.00329 (2019).
- 56 Toyama-Sorimachi, N. & Kobayashi, T. Lysosomal amino acid transporters as key players in inflammatory diseases. *Int Immunol* **33**, 853-858, doi:10.1093/intimm/dxab069 (2021).
- 57 Yambire, K. F. *et al.* Impaired lysosomal acidification triggers iron deficiency and inflammation in vivo. *Elife* **8**, doi:10.7554/eLife.51031 (2019).
- 58 Rawnsley, D. R. & Diwan, A. Lysosome impairment as a trigger for inflammation in obesity: The proof is in the fat. *EBioMedicine* **56**, 102824, doi:10.1016/j.ebiom.2020.102824 (2020).
- 59 Qian, M., Fang, X. & Wang, X. Autophagy and inflammation. *Clin Transl Med* **6**, 24, doi:10.1186/s40169-017-0154-5 (2017).
- 60 Bordon, Y. Immune regulation: lysosomes at the heart of inflammation. *Nat Rev Immunol* **11**, 502, doi:10.1038/nri3035 (2011).
- 61 Datta, G. *et al.* Endolysosome Localization of ERalpha Is Involved in the Protective Effect of 17alpha-Estradiol against HIV-1 gp120-Induced Neuronal

- Injury. *J Neurosci* **41**, 10365-10381, doi:10.1523/JNEUROSCI.1475-21.2021 (2021).
- 62 Van Acker, Z. P., Bretou, M. & Annaert, W. Endo-lysosomal dysregulations and late-onset Alzheimer's disease: impact of genetic risk factors. *Mol Neurodegener* **14**, 20, doi:10.1186/s13024-019-0323-7 (2019).
- 63 Hung, C. & Livesey, F. J. Endolysosome and autophagy dysfunction in Alzheimer disease. *Autophagy* **17**, 3882-3883, doi:10.1080/15548627.2021.1963630 (2021).
- 64 Muraleedharan, A. & Vanderperre, B. The Endo-lysosomal System in Parkinson's Disease: Expanding the Horizon. *J Mol Biol* **435**, 168140, doi:10.1016/j.jmb.2023.168140 (2023).
- 65 Todd, T. W., Shao, W., Zhang, Y. J. & Petrucelli, L. The endolysosomal pathway and ALS/FTD. *Trends Neurosci* **46**, 1025-1041, doi:10.1016/j.tins.2023.09.004 (2023).
- 66 Wendie A. Hasler, N. R., Xuesong Chen. in *HIV-Associated Neurocognitive Disorders* (ed Huangui Xiong Guoku Hu, Shilpa Buch) Ch. 17, 271-293 (Academic Press, 2024).
- 67 Qi, Z. *et al.* ROS-mediated lysosomal membrane permeabilization and autophagy inhibition regulate bleomycin-induced cellular senescence. *Autophagy* **20**, 2000-2016, doi:10.1080/15548627.2024.2353548 (2024).
- 68 Bae, M. *et al.* Activation of TRPML1 clears intraneuronal Abeta in preclinical models of HIV infection. *J Neurosci* **34**, 11485-11503, doi:10.1523/JNEUROSCI.0210-14.2014 (2014).
- 69 Datta, G. *et al.* Endolysosome localization of ERalpha is involved in the protective effect of 17alpha-estradiol against HIV-1 gp120-induced neuronal injury. *J Neurosci*, doi:10.1523/JNEUROSCI.1475-21.2021 (2021).
- 70 Hui, L. *et al.* Antiretroviral Drugs Promote Amyloidogenesis by De-Acidifying Endolysosomes. *J Neuroimmune Pharmacol* **16**, 159-168, doi:10.1007/s11481-019-09862-1 (2021).

April 23, 2025

RE: Life Science Alliance Manuscript #LSA-2025-03231-TR

Dr. Xuesong Chen
University of North Dakota
Biomedical Sciences
504 Hamline St
Grand Forks, ND 58203

Dear Dr. Chen,

Thank you for submitting your revised manuscript entitled "SLC38A9 is directly involved in Tat-induced endolysosomal dysfunction and senescence in astrocytes". As you will see, all reviewers are satisfied and recommend publication. We would be happy to publish your paper in Life Science Alliance pending final revisions necessary to meet our formatting guidelines.

- please be sure that the authorship listing and order is correct.
- please add a Category for your manuscript in our system.
- please add ORCID ID for secondary corresponding author -- they should have received instructions on how to do so.
- please indicate both corresponding authors in your manuscript.
- please upload clean manuscript file without highlights.
- please add the X and Bluesky handles of your host institute/organization as well as your own or/and one of the authors in our system.
- please add scale bars to the images in Figure 1, and ensure scale bars in 2B and 4C are legible.

A. FINAL FILES:

B. MANUSCRIPT ORGANIZATION AND FORMATTING:

per figure for this information. These files will be linked as supplementary "Source Data" files.

Sincerely,

Reviewer #1 (Comments to the Authors (Required)):

In this revised manuscript, the authors performed a couple of new experiments as suggested, while the remaining comments were addressed by providing literature references.

Reviewer #2 (Comments to the Authors (Required)):

Only one minor issue about the manuscript's title was identified. In response, the authors revised the title as suggested. There are no other comments.

Reviewer #3 (Comments to the Authors (Required)):

All comments have been adequately addressed.

April 28, 2025

RE: Life Science Alliance Manuscript #LSA-2025-03231-TRR

Dr. Xuesong Chen
University of North Dakota
Biomedical Sciences
504 Hamline St
Grand Forks, ND 58203

Dear Dr. Chen,

Thank you for submitting your Research Article entitled "SLC38A9 is directly involved in Tat-induced endolysosomal dysfunction and senescence in astrocytes". It is a pleasure to let you know that your manuscript is now accepted for publication in Life Science Alliance. Congratulations on this interesting work.

DISTRIBUTION OF MATERIALS:

Again, congratulations on a very nice paper. I hope you found the review process to be constructive and are pleased with how the manuscript was handled editorially. We look forward to future exciting submissions from your lab.

Sincerely,
